# Screening macrophage polarization genes in spinal cord injury as therapeutic targets

Xiaowei Zha[1], Shen Cao[2]*

**1** Medical Faculty Heidelberg, Heidelberg University, Heidelberg, Germany, **2** Department of Orthopedics, Anhui No.2 Provincial People's Hospital, Hefei, China

\* 15025113@vnu.edu.vn

## Abstract

Macrophage polarization correlates strongly with the progression and prognosis of spinal cord injury (SCI), yet the therapeutic potential of macrophage polarization-related genes (MPRGs) in SCI remains unexplored. This study identified hub genes associated with MPRGs for SCI diagnosis, prognosis, and therapy. Differentially expressed genes (DEGs) between SCI and control groups were intersected with MPRGs to identify six differentially expressed MPRGs (DE-MPRGs). Machine learning algorithms, including LASSO, RF, and XGBoost, selected three hub genes (*Soat1*, *Comt*, and *Myo1f*) with elevated expression in SCI samples. Functional enrichment analysis indicated involvement in immune-related pathways. Immune infiltration analysis revealed differences in 11 immune gene sets between SCI and controls, all positively correlated with the hub genes. In silico drug prediction identified 37 small molecules, including dexamethasone and atorvastatin, as potential modulators of macrophage polarization in SCI. Single-cell RNA sequencing showed significantly higher expression of all hub genes in M2 than in M1 macrophages. RT-qPCR validation confirmed upregulation of the hub genes in SCI models. These results highlight *Soat1*, *Comt*, and *Myo1f* as novel hub genes in SCI, offering insights into macrophage polarization mechanisms and potential therapeutic targets.

## 1. Introduction

Spinal cord injury (SCI) emerges as a critical neurological disorder from diverse traumatic episodes, inducing either partial or total disruption of neural operations [1]. Chief origins of SCI involve injury-provoking incidents, including falls, sports-induced harms, vehicle crashes, and slips, alongside non-injury-based contributors such as tumors and intervertebral disc disorders [2–4]. Projections worldwide reveal that exceeding two million persons endure the impacts of SCI [5–7]. In addition to the corporeal and emotional agony induced by SCI, the consequent complications exert a considerable economic and societal strain upon households and populations [5]. Although therapeutic modalities for SCI have progressed, exemplified by stem cell

**Data availability statement:** All gene expression data used in this study were obtained from the Gene Expression Omnibus (GEO) database, which is hosted by the National Center for Biotechnology Information (NCBI; https://www.ncbi.nlm.nih.gov/geo/). The unique accession numbers for the relevant datasets are as follows: GSE45550, GSE45006, GSE183591, and GSE213240. All datasets are freely accessible without additional authorization via the search function on the GEO database official website (by entering the corresponding accession number). No new original datasets were generated in this study. Additional processed data and analysis outputs related to this work are provided in the Supporting Information.

**Funding:** This work was supported by the General Programs of Natural Science Research of Anhui Provincial Education Department (Grant No. ZR2022B001). The funders had no role in study design, data collection and analysis, decision to publish, or preparation of the manuscript.

**Competing interests:** The authors have declared that no competing interests exist.

engraftment, viable approaches to neural regeneration continue to be scarce. Investigations in modern medicine strive to innovate interventions for spinal cord restoration, demanding a thorough comprehension of the molecular and neural pathways implicated in SCI development.

Following SCI, macrophages, key components of the inflammatory response, play a pivotal role in driving inflammatory processes [8–10]. After SCI, macrophage phenotypes undergo dynamic changes that may influence inflammatory progression, glial and fibrous scar formation, and spinal cord regeneration [11]. Macrophages have traditionally been categorized into two phenotypes: M1 and M2 [12]. M1 macrophages (classically activated macrophages) are the predominant phenotype at the injury site and secrete high levels of pro-inflammatory cytokines in response to stimulation from the injured microenvironment. These pro-inflammatory cytokines induce neurotoxicity, triggering the death of neural cells and impeding spinal cord regeneration [13]. In contrast, M2 macrophages (alternatively activated macrophages) attenuate inflammatory escalation and exert a neuroprotective effect by releasing anti-inflammatory factors and neurotrophic molecules, stimulating angiogenesis, and promoting neuronal and tissue regeneration [14]. Numerous studies have shown that promoting macrophage polarization from the M1 to the M2 phenotype enhances neuronal repair and motor function recovery while attenuating secondary injury in SCI [15]. Given the substantial influence of macrophage polarization on disease progression and prognosis, increasing attention has been directed toward macrophage polarization-related genes (MPRGs) [10]. MPRGs also participate in multiple biological processes, including immune regulation, metabolic homeostasis, autophagy, apoptosis, pathogen clearance, and neurological modulation [16–18]. Owing to their diverse roles in the immune system, regulating the upstream and downstream pathways of MPRGs is essential for maintaining immune homeostasis, modulating inflammatory responses, facilitating tissue repair, and supporting other physiological functions. Therefore, MPRGs may serve as potential therapeutic targets for SCI.

Accordingly, this study aims to identify hub genes among MPRGs with potential diagnostic and therapeutic value in SCI, thereby providing a theoretical basis for elucidating SCI pathogenesis and informing clinical interventions.

## 2. Materials and methods

### 2.1. Data source and preprocessing

Data sets associated with SCI, namely GSE213240, GSE183591, GSE45006, and GSE45550, have been obtained from the Gene Expression Omnibus (GEO) repository (https://www.ncbi.nlm.nih.gov/geo/ [Accessed: March 15, 2023]). In particular, the GSE45550 collection, originating from *Rattus norvegicus* soleus muscle tissue, incorporates 18 specimens from SCI cases (at 3, 8, and 14 days following injury) along with 6 control specimens [19]. For GSE45006, isolation occurred from *Rattus norvegicus* spinal cord tissue, yielding 20 SCI specimens and 4 sham-operated controls across time points of 1 day, 3 days, 1 week, 2 weeks, and 8 weeks after injury. Similarly, GSE183591, drawn from Sprague Dawley (SD) rat spinal cord, features 16 SCI specimens and 4 sham-operated controls evaluated at 1 week, 2 weeks, 4 weeks,

and 8 weeks post-injury [20]. The GSE213240 single-cell RNA sequencing (scRNA-seq) dataset (platform: GPL25947) consists of 8 spinal cord tissue samples from SCI patients, sequenced via high-throughput technology. Expression matrices, sample clinical information, and annotation files for the corresponding microarray platforms were obtained using the GEOquery package. Probe ID-to-gene name mappings were extracted from the annotation files, and probe IDs in the expression matrices were converted to gene names by merging the data. Gene name cleaning was then performed by splitting and removing redundant entries separated by spaces, followed by deduplication to retain unique expression records for each gene. Sample IDs and phenotypic data were extracted from the clinical information to construct group files, and the processed expression matrices and group information were saved as standardized files. To minimize biases due to small sample sizes, the "sva" R package was used to integrate the GSE45550 and GSE45006 datasets, which were subsequently designated as the training set. Batch effects were corrected using the ComBat algorithm (based on an empirical Bayesian framework) prior to merging, and post-merge principal component analysis (PCA) validated the complete removal of batch clustering (S1A, B Fig). GSE183591 served as the external validation set. Additionally, a total of 35 macrophage polarization-related genes (MPRGs) were retrieved from the published literature [21]. Three M1 macrophage marker genes (*TNF-α*, *HLA-DRα*, and *iNOS*) and five M2 macrophage marker genes (*IL-10*, *CCL22*, *ARG-1*, *CD206*, and *CD163*) were retrieved from published literature [22].

## 2.2. Identification of differentially expressed genes (DEGs) between SCI and control groups

After batch effect correction, DEGs distinguishing SCI cases from controls within the GSE45006 and GSE45550 datasets were detected via the "limma" R package (version 3.54.0), under thresholds of adjusted $P < 0.05$ and $|\log_2FC| > 0.5$ [23]. Enrichment analysis for the biological functions of these DEGs (adjusted $P < 0.05$) was then conducted, employing the "clusterProfiler" R package (version 4.7.1) in conjunction with the "org.Rn.e.g.,db" annotation package, and leveraging resources from the Kyoto Encyclopedia of Genes and Genomes (KEGG, https://www.genome.jp/kegg/ [Accessed: March 20, 2023]) as well as Gene Ontology (GO, https://www.geneontology.org/ [Accessed: March 20, 2023]) databases [24,25].

## 2.3. Screening of the hub genes

DE-MPRGs were initially ascertained via the overlap between DEGs and MPRGs, computed with the aid of the "ggvenn" R package (version 0.1.9). Thereafter, selection of genes demonstrating substantial biological relevance was accomplished through application of extreme gradient boosting (XGBoost), random forest (RF), and least absolute shrinkage and selection operator (LASSO) techniques. Specifically, for the LASSO model, family = "binomial" was set to suit binary classification tasks, and type.measure = "auc" was designated as the performance evaluation metric. A regularization parameter ($\lambda$) was introduced, and lambda.min (the value yielding the minimum error) was selected as the optimal regularization parameter via 5-fold cross-validation, followed by extraction of genes with non-zero coefficients. For the RF model, optimal parameters were set as mtry = 3 and ntree = 1,000; ensemble learning across multiple decision trees reduced the variance of individual models, while built-in out-of-bag (OOB) sample validation was used instead of an independent test set to evaluate model generalization without additional sample consumption. For the XGBoost model, a reasonable learning rate (eta = 0.5), tree depth (max_depth = 5), and number of iterations (nround = 25) were configured to balance fitting ability and complexity, with gradient boosting mechanisms employed to iteratively correct errors. Hub genes were designated as those emerging from the common overlap among the three machine learning algorithms. To eliminate the potential impact of data pooling, DEGs of GSE45006 and GSE45550 were separately identified using the "limma" R package (version 3.54.0) (adjusted $P < 0.05$ and $|\log_2FC| > 0.5$). Subsequently, the "RobustRankAggreg" (RRA) package (version 3.54.0) was applied to integrate the ranking of DEGs across the two datasets. The aggregateRanks function was utilized to calculate the integrated score for each gene, with a lower score indicating a higher ranking. This analysis generated a cross-dataset list of significantly integrated genes, and special attention was paid to the rankings of the core genes within this integrated list.

After that, correlations between these hub genes were investigated through Spearman correlation analysis. Employing the GENEMANIA database (http://genemania.org/ [Accessed: April 27, 2023]), an interaction network linking genes for the hub genes was established. Ultimately, incorporation of the hub genes into a nomogram was achieved via the "rms" R package (version 6.2.0). Reliability of this nomogram was assessed by means of receiver operating characteristic (ROC) curves and decision curve analysis (DCA).

## 2.4. Gene set enrichment analysis (GSEA) of the hub genes

Training set specimens were categorized into cohorts displaying elevated and reduced expression utilizing the median values from hub genes, prompting subsequent differential expression evaluation. Genes in the entire collection received ordering according to $\log_2$FC metrics. Application of the "ClusterProfiler" R package (version 4.7.1) enabled GSEA implementation under the cutoff of adjusted $P < 0.05$ [26].

## 2.5. Ingenuity pathway analysis (IPA) of the hub genes

To explore the biological functions of the hub genes, IPA was performed based on the QIAGEN Knowledge Base (https://digitalinsights.qiagen.com/products-overview/qiagen-knowledge-base/ [Accessed: April 16, 2023]). Subsequently, we analyzed the diseases and functional annotations to which the hub genes are enriched.

## 2.6. Immune analysis

Gene sets pertaining to immune responses were sourced from the ImmPort repository (https://www.immport.org/ [Accessed: April 21, 2023]). Through single-sample gene set enrichment analysis (ssGSEA) executed within the "GSVA" R package (version 1.46.0), enrichment metrics for such gene sets were derived for control and SCI cohorts [27]. Variances in these enrichment metrics across the control and SCI cohorts underwent evaluation. Associations linking differentially expressed immune response gene sets to hub genes were probed via Spearman's rank correlation technique. Moreover, marker genes indicative of human M2 and M1 macrophages were mapped to their rat orthologs employing the "homologene" R package. Scrutiny extended to the relationships connecting hub genes with these rat M1/M2 macrophage marker genes (stemming from human orthologs).

## 2.7. Evaluation of immune infiltration

Upon procurement of gene expression profiles, immune cell infiltration abundances were estimated employing the CIBER-SORT method. Relative abundances among 22 immune cell subtypes per specimen were illustrated through stacked bar charts, alongside evaluation of infiltration variances across cohorts. Moreover, associations linking differentially infiltrated immune cell subtypes to one another, plus those connecting hub genes to such subtypes, were examined via Spearman's rank correlation.

## 2.8. Creation of micro RNA (miRNAs)-mRNA network and transcription factors (TFs)-mRNA network

Orthologs corresponding to hub genes in humans were mapped through the "homologene" R package. miRNet repository (https://www.mirnet.ca/miRNet/home.xhtml [Accessed: April 29, 2023]) facilitated the forecasting of miRNAs directed toward hub genes. Concurrently, transcription factors (TFs) aimed at these genes underwent prediction leveraging the NetworkAnalyst tool (https://www.networkanalyst.ca/ [Accessed: April 30, 2023]). Interactions in TF-mRNA and miRNA-mRNA regulatory frameworks were depicted employing Cytoscape software (version 3.8.0) [28].

## 2.9. Drug prediction of the hub genes

Conversion to human orthologs for hub genes was executed by means of the "homologene" R package. Queries in the Drug-Gene Interaction Database (DGIdb; https://dgidb.genome.wustl.edu/ [Accessed: April 27, 2023]) relied on individual hub genes functioning as search keywords to pinpoint pharmaceuticals that engage with them.

## 2.10. Processing and annotation of scRNA-seq data

Analysis at the single-cell resolution scrutinized the interplay linking hub genes to macrophage polarization. Sequencing profiles derived from GSE213240 initially underwent filtration through the "Seurat" R package (version 5.0.1) [29]. In detail, quality assurance imposed three thresholds: mitochondrial gene percentage confined below 10%, detected features per cell (nFeature) bounded from 200 to 2,500, and counts per cell (nCount) capped under 6,000. Thereafter, selection of the foremost 2,000 highly variable genes occurred via the FindVariableFeatures tool, relying on the association of expression averages with dispersions; these selections persisted for ongoing evaluations. Dataset specimens subsequently experienced PCA. The JackStraw function was employed to conduct a permutation test for the null distribution, aiming to quantify the percentage of variance explained by the top 30 principal components (PCs). Prominent variance-explaining principal components (PCs) underwent selection through examination of the elbow curve diagram, whereupon cellular unsupervised grouping ensued via invocation of FindClusters and FindNeighbors functions (resolution = 0.2). Labeling and contrasting of cell identities proceeded with deployment of the "SingleR" R package (version 2.0.0) integrated alongside the CellMarker repository (http://117.50.127.228/CellMarker/ [Accessed: April 22, 2023]). Ultimately, disparities in hub gene expression across M2 and M1 macrophage populations received scrutiny (adjusted $P < 0.05$; Bonferroni correction).

## 2.11. Expression analysis of the hub genes

For additional corroboration of the robustness inherent in our results, disparities in hub gene expression across control and SCI cohorts were assessed via the Wilcoxon rank-sum test ($P < 0.05$) within the training cohort together with the GSE183591 collection.

## 2.12. Spinal cord injury rat model

Female Wistar rats in good health (*Rattus norvegicus*; 200 g body weight, aged 8 weeks, sourced from Janvier Labs, France) underwent random allocation into one sham-operated group and three SCI groups (3 days, 7 days, and 14 days post-injury). The initial number of animals in each SCI group was 8 rats, whereas the sham-operated group consisted of 7 rats. Compliance with the NIH Guidelines for the Care and Use of Laboratory Animals (8th edition, revised 2010, NIH) guided the investigation, wherein the fewest possible animals per cohort sufficed to yield data robust enough for statistical inference, aligning with core ethical tenets; experimental designs secured endorsement from the Anhui Medical University Animal Ethics Committee (approval LLSC20231977, Hefei, China), while every operation conformed to prevailing ethical norms and followed the ARRIVE Checklist (S1 File). Proper husbandry attended to all subjects, who occupied enclosures limited to at most three individuals apiece, maintained within specific pathogen-free (SPF) environments featuring regulated ambient temperature (22 ± 1 °C), humidity levels (50 ± 1%), alongside a 12 h light/12 h dark photoperiod. Ad libitum access prevailed for both feed and hydration.

As indicated beforehand, induction of the contusion/compression SCI paradigm relied on a customized 28 g aneurysm clip (Fehlings Laboratory, Canada). Concisely, subjects received a 1:1 blend of $N_2O$ and $O_2$ alongside isoflurane (1.5–3%) for eliciting systemic anesthesia. Subsequently, posterior integument incision occurred at the T10–T8 cord segment, trailed by T9 lamina resection to unveil the neural tissue; the clip then encircled the cord and persisted for 60 s, culminating in a compression-contusion lesion [30–32]. Rats, upon emergence from anesthetic effects, exhibited trailing of hindlimbs. Laminotomy at T9, sparing neural integrity, constituted the sham cohort. Application of ocular salve averted corneal aridity, whereas a thermal mat preserved subject thermoregulation. The animals received comprehensive post-surgical care. Immediately following surgery, they were administered a subcutaneous injection of 4 ml saline. The bladder was manually emptied two to three times daily (based on signs of wet abdomen and bladder distension) until spontaneous urination was re-established. Preventive antibiotics, including 2.5% Baytril (enrofloxacin, Bayer AG, Leverkusen, Germany), meloxicam (2.0 mg/kg), and buprenorphine (0.05 mg/kg subcutaneously), were administered daily. All analgesics

and antibiotics were administered daily after surgery and continued for up to seven consecutive days, or until the day of sacrifice for animals in the 3-day group. Treatment was discontinued once rats regained spontaneous urination and showed no obvious signs of pain or infection. Throughout the experiment, trained personnel monitored the animals daily for signs of pain, stress, or other discomfort. The animals' general condition was assessed using a scoring system, and appropriate interventions were performed based on the scores to minimize suffering. If the predetermined humane end-point criteria were met, the animals were euthanized immediately. Euthanasia was performed by placing the animals in an induction chamber and administering 5% isoflurane by inhalation until they reached a state of complete unconsciousness, as indicated by the absence of righting reflex, respiration, and response to noxious stimuli. Once unconsciousness was confirmed, subsequent experimental procedures were conducted. Animals from the experimental group were sacrificed on days 3, 7, and 14 post-SCI for spinal cord tissue collection. A total of 7 rats per group were included in the subsequent experiments (1 rat died in each SCI group during surgery or the experimental period, while all 7 rats in the sham-operated group survived). This approach ensured that the animals remained unaware during euthanasia and complied with internationally recognized guidelines, including the AVMA Guidelines for the Euthanasia of Animals (2020 Edition) and Directive 2010/63/EU of the European Parliament. Additional information is provided in the Supporting Information, including the Inclusivity in global research questionnaire (S2 File) and the PLOS ONE humane endpoints checklist (S3 File).

## 2.13. Reverse transcription-quantitative PCR (RT-qPCR)

T9 spinal cords were collected at 3, 7, and 14 days post-SCI or after laminectomy, immediately frozen, and stored at −80°C. Total RNA was then extracted using the NucleoSpin® RNA Isolation Kit (Macherey-Nagel GmbH & Co., Düren, Germany). Primers for each gene are listed in Table 1.

Relative expression levels of specific genes/markers in RT-qPCR were assessed using the Luna® Universal RT-qPCR Master Mix system (New England Biolabs, Massachusetts, USA). RT-qPCR was performed using the StepOne Plus system (Applied Biosystems, Massachusetts, USA). Duplicate cycle thresholds (CTs) were averaged, and data were normalized to the expression of β-actin mRNA using the $2^{-\Delta\Delta Ct}$ method [33].

## 2.14. Statistical analysis

R software (https://www.r-project.org/) enabled the implementation of all statistical computations. Disparities across cohorts received evaluation through Wilcoxon scrutiny, signifying meaningful divergence when $P < 0.05$. For RT-qPCR evaluations, Gaussian-form data invoked Student's t-test application, while skewed distributions prompted Wilcoxon rank-sum deployment, with thresholds at $P < 0.05$ marking evidentiary weight. Multiplicity safeguards, where applicable, entailed P-value recalibration via the Benjamini-Hochberg (BH) procedure to mitigate false discovery rate (FDR).

**Table 1. Primers for RT-qPCR.**

| Gene | Primers forward (5'-3') |
|---|---|
| *Myo1f* | Forward: TCAACCGGAACTTTGTCGGG |
| | Reverse: AGTCCCGTTTGATGGGCTTG |
| *Soat1* | Forward: AGCCAAAGATCTGAGAGCACC |
| | Reverse: GAACTCAAGGACCAGCCTTCC |
| *Comt* | Forward: GGCCTTGAGGAGATGCCGT |
| | Reverse: GATGCGCTGCTCCTTTGTGT |
| *β-actin* | Forward: AACCTTCTTGCAGCTCCTCCG |
| | Reverse: ATACCCACCATCACACCCTGG |

# 3. Results

## 3.1. Acquisition and enrichment analysis of DEGs between SCI and control groups

Between SCI and control cohorts, a cohort of 1,396 DEGs surfaced, featuring 820 instances of upregulation coupled with 576 cases of downregulation (Fig 1A; S1 Table). Hierarchical clustering via heatmap illustrated the premier 20 upregulated alongside downregulated genes (Fig 1B). Herein exhibited stand the foremost 10 enriched designations from GO alongside routes from KEGG. Across 1,297 GO designations, DEGs manifested enrichment (S2 Table), partitioned into 1,094 entries for biological process (BP), 137 for cellular component (CC), and 66 for molecular function (MF). As illustrations, engagement by DEGs encompassed membrane microdomain, membrane raft, synaptic membrane, positive regulation of response to external stimulus, immune response-regulating signaling pathway, myeloid leukocyte activation, glycosaminoglycan binding, cell adhesion molecule binding, and proteoglycan binding (Fig 1C). Among the KEGG pathways, 126 were significantly enriched (S3 Table), such as osteoclast differentiation, NF-κB signaling pathway, and pertussis (Fig 1D).

## 3.2. Acquisition of the hub genes

Six differentially expressed macrophage polarization-related genes (DE-MPRGs) were identified by intersecting DEGs with MPRGs: *Fgd2*, *Soat1*, *Comt*, *Myo1f, Maf*, and *Sh2b2* (Fig 2A). The least absolute shrinkage and selection operator (LASSO) algorithm selected three characteristic genes: *Myo1f*, *Soat1*, and *Comt* (Fig 2B). The extreme gradient boosting (XGBoost) algorithm identified five characteristic genes: *Fgd2*, *Soat1*, *Comt*, *Myo1f*, and *Sh2b2* (Fig 2C). The random forest (RF) algorithm identified four characteristic genes: *Fgd2*, *Soat1*, *Comt*, and *Myo1f* (Fig 2D-E). The intersection of characteristic genes from the three algorithms yielded three hub genes: *Soat1*, *Comt*, and *Myo1f* (Fig 2F). Meta-analysis using the RRA package revealed that three genes were stably identified in the integrated results. Specifically, *Soat1* was ranked 1,336th (top 8.84%, score = 0.0936), *Myo1f* was ranked 1,605th (top 10.62%, score = 0.1139), and *Comt* was ranked 2,972nd (top 19.66%, score = 0.2258) (S4 Table). Correlation analysis revealed significant positive correlations among the hub genes (Fig 2G), with the strongest correlation between *Soat1* and *Myo1f* (R = 0.901, *P* = 5.77e-18) (Fig 2H). To explore interactions among the three hub genes, a gene-gene interaction network was constructed (Fig 2I). These genes formed a complex interaction network, with the following interaction types: co-expression (39.02%), physical interactions (25.61%), predicted interactions (21.99%), pathway associations (8.15%), co-localization (2.86%), and shared protein domains (2.37%). A nomogram incorporating the three hub genes was developed to predict SCI progression (Fig 3A). The nomogram's ROC curve yielded an area under the curve (AUC) of 0.866 (Fig 3B). The DCA curve demonstrated that the nomogram provided a higher net benefit (Fig 3C). These findings indicate that the nomogram exhibits high predictive accuracy for SCI progression.

## 3.3. Enrichment analysis of the hub genes

To further elucidate the roles of hub genes in SCI pathogenesis, GSEA was performed. The top five enriched KEGG pathways are presented. In the high-expression group, all hub genes were enriched in natural killer cell-mediated cytotoxicity, systemic lupus erythematosus, lysosome, and complement and coagulation cascades (Fig 4A-C). Additionally, both *Soat1* and *Myo1f* were enriched in the cytokine-cytokine receptor interaction pathway in the high-expression group (Fig 4B-C).

## 3.4. IPA of the hub genes

A total of 582 canonical pathways were significantly enriched for DEGs between the SCI and control groups, with the top 20 enriched pathways presented (Fig 5A). The three hub genes were involved in four canonical pathways: dopamine receptor signaling, L-DOPA degradation, noradrenaline and adrenaline degradation, and dopamine degradation. Among these, the dopamine receptor signaling pathway had the highest |z-score| and is therefore illustrated in Fig 5B. In terms

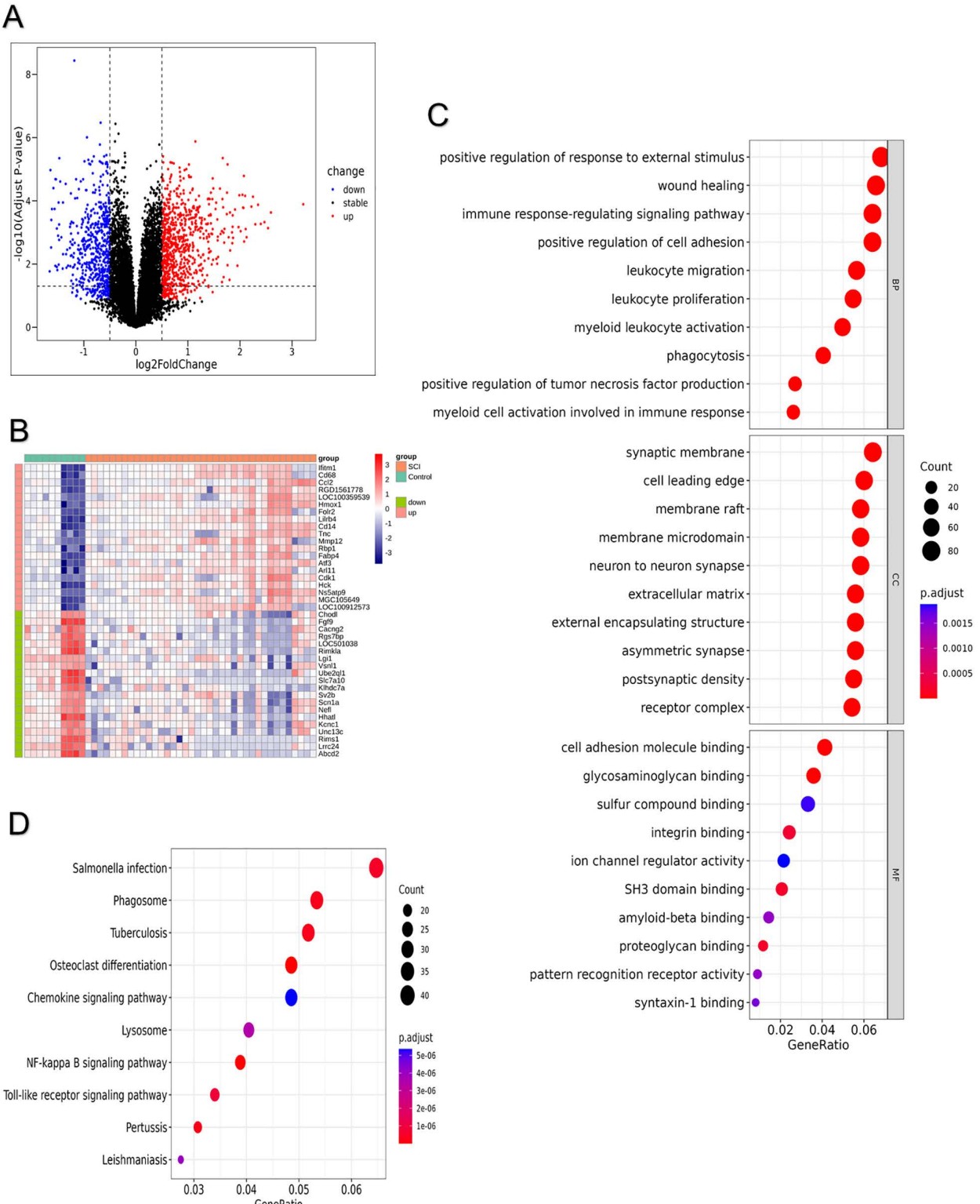

**Fig 1. Differentially expressed genes analysis and enrichment analysis of DEGs. (A)** Volcano plot of DEGs. **(B)** Heatmap of differentially expressed genes (Top 20 DEGs). **(C)** GO enrichment analysis of DEGs. **(D)** KEGG enrichment analysis of DEGs.

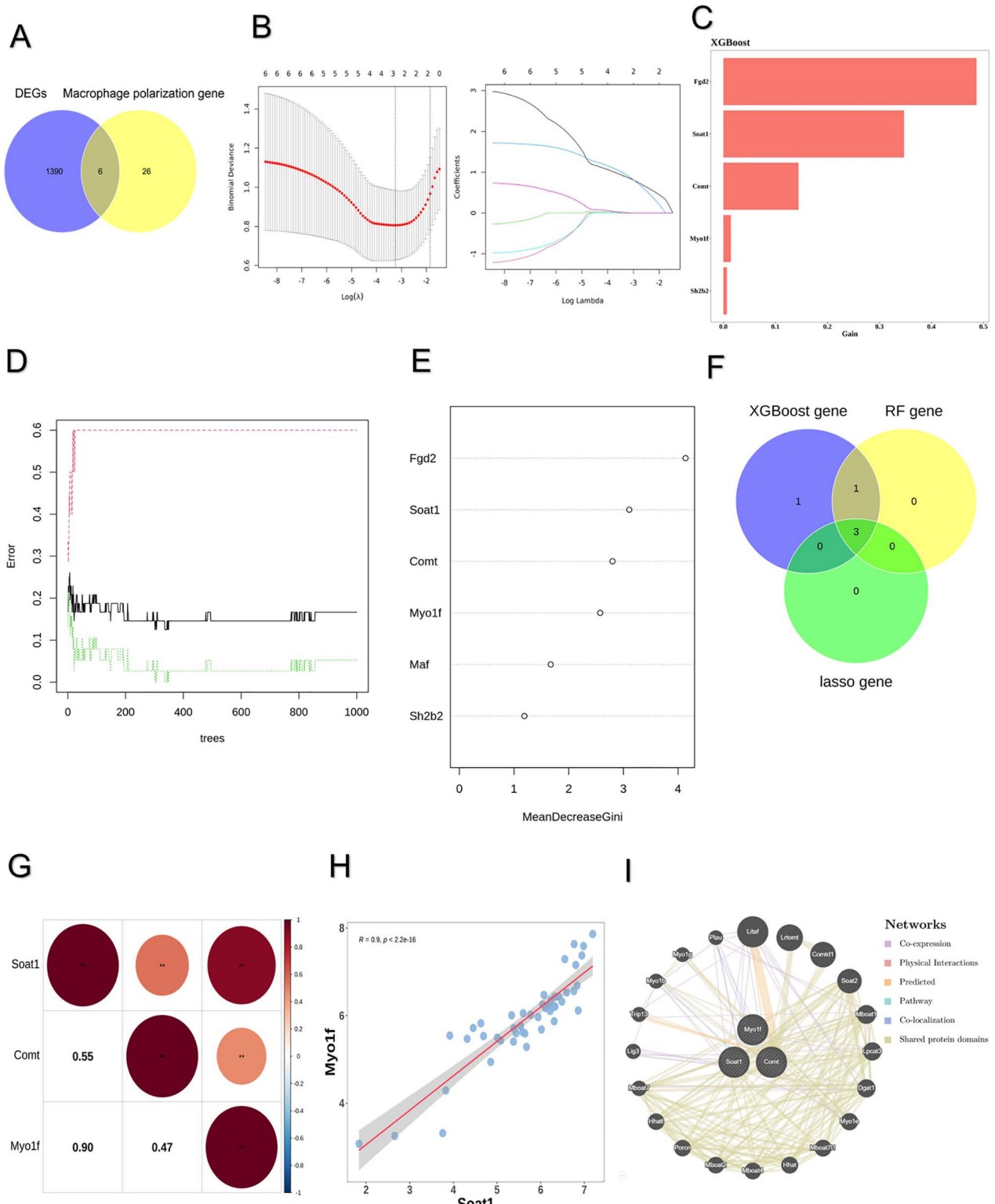

**Fig 2. Acquisition of the hub genes. (A)** Venn plot of DEGs and macrophage polarization related genes (MPRGs). **(B)** LASSO algorithm. **(C)** XGBoost algorithm. **(D)** The number of RF trees correlates with model errors. **(E)** Average minimum Gini coefficient diagram. **(F)** Venn plot of three machine learning. **(G)** Hub gene correlation analysis. **(H)** Scatter plots of the two genes with the highest positive correlation. **(I)** PPI protein interaction network.

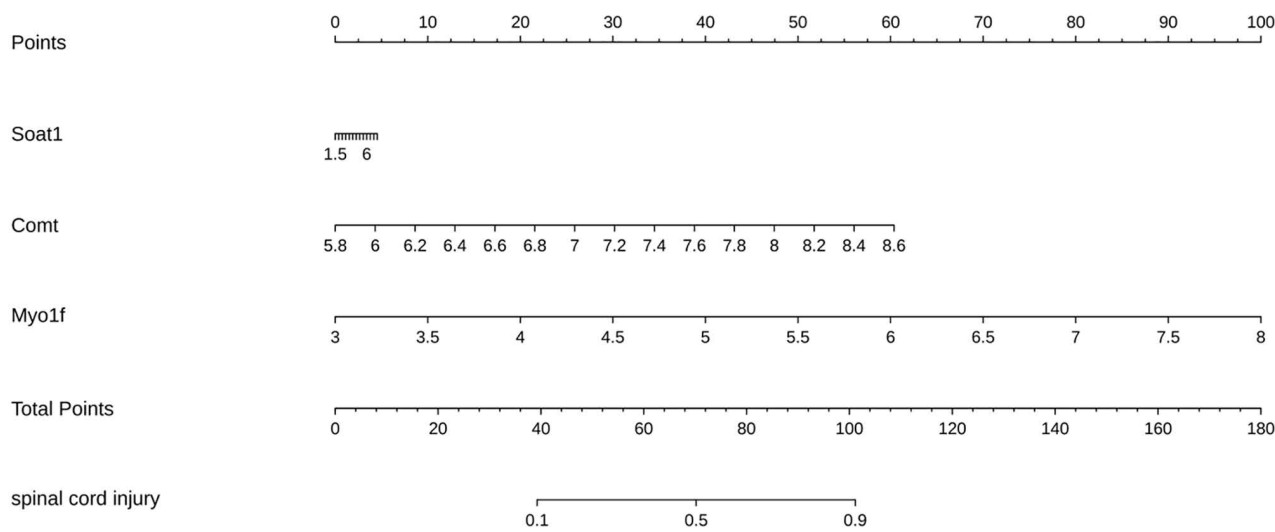

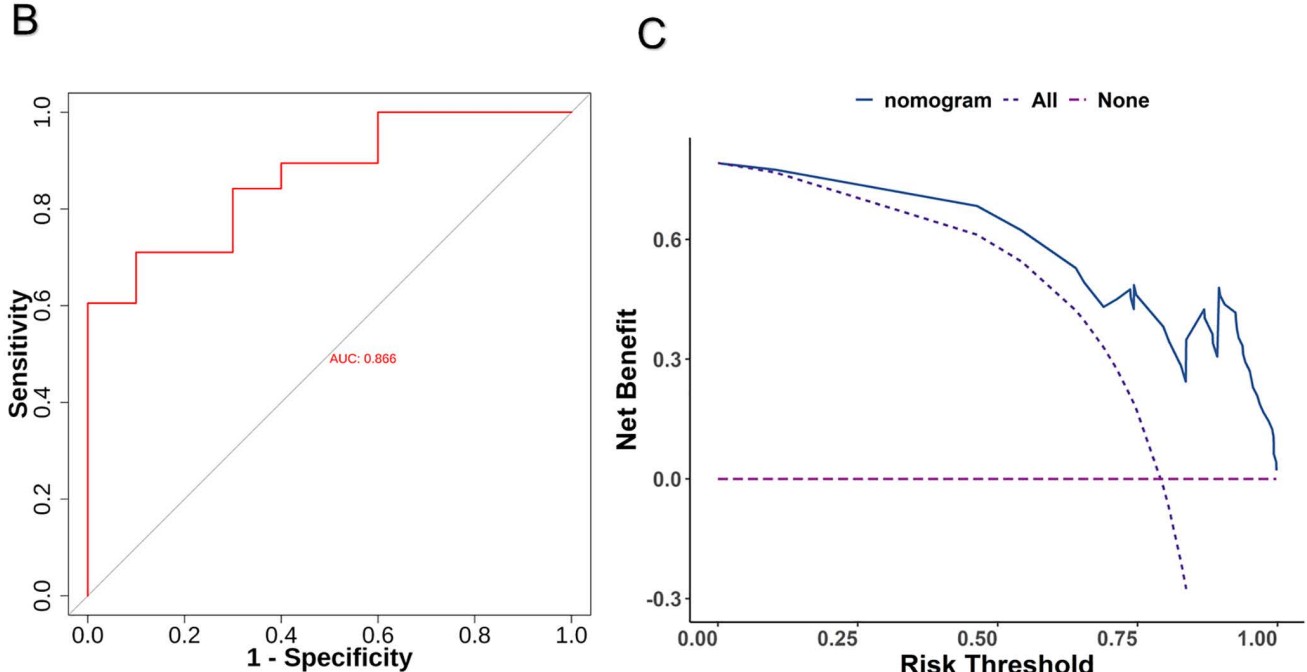

**Fig 3. Construction and evaluation of nomogram model. (A)** The nomogram containing the three hub genes. **(B)** ROC curve. **(C)** DCA curve.

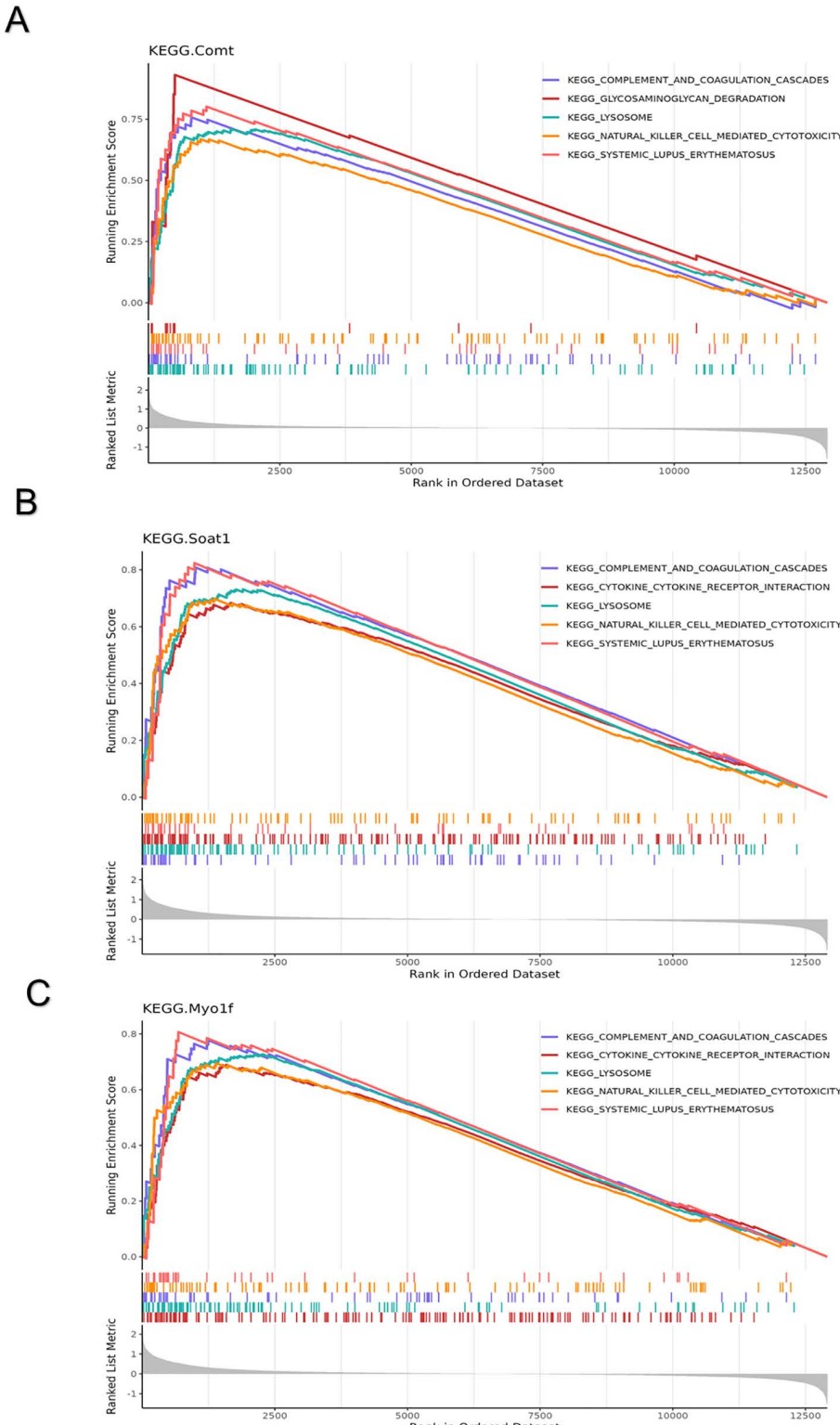

**Fig 4. GSEA analysis of the hub genes. (A)** GSEA analysis of *Comt*. **(B)** GSEA analysis of *Soat1*. **(C)** GSEA analysis of *Myo1f*.

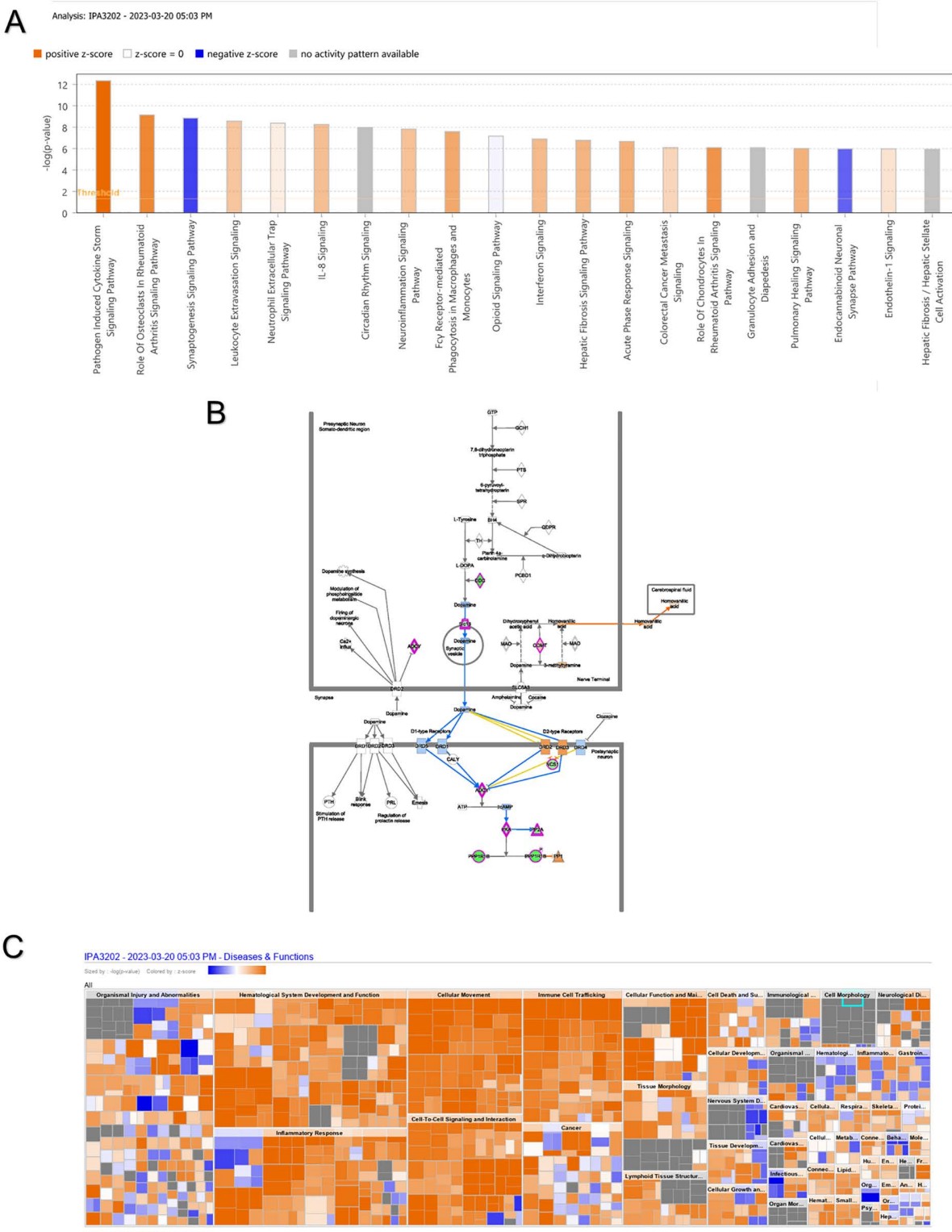

**Fig 5. IPA of the hub genes. (A)** Top 20 classical IPA pathways **(B)** Interactive mapping of dopamine receptor signaling pathways. **(C)** Statistical analysis of disease and functional enrichment.

of disease and functional annotations, the DEGs were primarily associated with organismal injury and abnormalities (Fig 5C).

### 3.5. Immune analysis between SCI and control groups

In SCI cohorts compared to controls, eleven gene sets associated with immune responses displayed substantial differential expression, spanning tumor necrosis factor (TNF) family member receptors, T cell receptor (TCR) signaling pathway, transforming growth factor-beta (TGF-β) family member receptors, interferon receptors, natural killer cells, interleukin receptors, B cell receptor (BCR) signaling pathway, cytokine receptors, antimicrobials, chemokine receptors, and antigen processing and presentation (Fig 6A; S5 Table). Scrutiny of correlations uncovered strong positive linkages across all hub genes and every differentially expressed immune response gene set, with the peak association evident between *Myo1f* and the B cell receptor (BCR) signaling pathway (R=0.925, *P*=5.44e-22) (Fig 6B). The marker genes of M1 and M2 macrophages from humans were converted to their orthologous genes in rats, namely *RT1-Da*, *Tnf*, *Cd163*, *Mrc1*, *Arg1*, *Ccl22*, and *Il10*. The relevant analysis indicated that *Mrc1* was markedly positively associated with all hub genes (Fig 6C). *RT1-Da* and *Cd163* were markedly positively associated with *Myo1f* and *Soat1*. Additionally, *Comt* was negatively associated with *Tnf* and *Ccl22*.

### 3.6. Analysis of immune cell infiltration

The CIBERSORT algorithm facilitated quantification of infiltration fractions for 22 immune cell varieties within specimens, underscoring the pivotal influence of alterations in the immune milieu and associated signaling cascades on SCI advancement and recovery (Fig 7A). Elevated abundances of CD8$^+$T cells and resting dendritic cells marked the SCI cohort relative to controls, as our observations revealed. In opposition, the SCI ensemble manifested reduced incidences of memory CD4$^+$T cells, M0 macrophages, and follicular helper T cells (Tfh cells) when juxtaposed against the control ensemble (Fig 7B). Associations among immune infiltrates in these specimens underwent further interrogation (Fig 7C). Plasma cells forged a marked affirmative bond with resting natural killer (NK) cells (R=0.63), while memory B cells evinced pronounced discord with naive B cells (R=−0.83). Associations spanning the triad of hub genes and immune infiltrates received additional examination (Fig 7D). Inverse linkages tied the three hub genes to resting memory CD4$^+$T cells, follicular helper T cells, M0 macrophages, and plasma cells. Affirmative ties linked every one of the three hub genes to resting dendritic cells and CD8$^+$T cells, mirroring the intergroup disparities noted earlier.

### 3.7. The TF-mRNA-miRNA networks

Elucidation of the mechanistic underpinnings dictating hub gene operations in SCI necessitated assembly of regulatory architectures encompassing miRNA-mRNA and TF-mRNA interactions. The miRNA-mRNA construct, encompassing 273 miRNAs alongside 3 mRNAs (specifically, the set of three hub genes), emerged (Fig 8A). Illustrative regulatory pairings spanned *MYO1F*-hsa-miR-27b-3p, *SOAT1*-hsa-miR-485-5p, *COMT*-hsa-miR-637, and similar associations. Complementing this, the TF-mRNA framework integrated 30 TFs with the 3 mRNAs (i.e., the three hub genes) (Fig 8B). For instance, the TF TEAD1 could regulate *SOAT1* and *COMT*. Three TFs (TFAP2C, NFKB1, and CREB1) were common TFs regulating both *COMT* and *MYO1F*.

### 3.8. Drug prediction

The hub gene-drug interaction network was constructed using DGIdb. The results revealed that a total of 37 small-molecule drugs were predicted by the database (Fig 8C). Five small-molecule drugs were associated with *SOAT1*: nifedipine, dexamethasone, testosterone, atorvastatin, and lovastatin. For *COMT*, 32 small-molecule drugs were predicted,

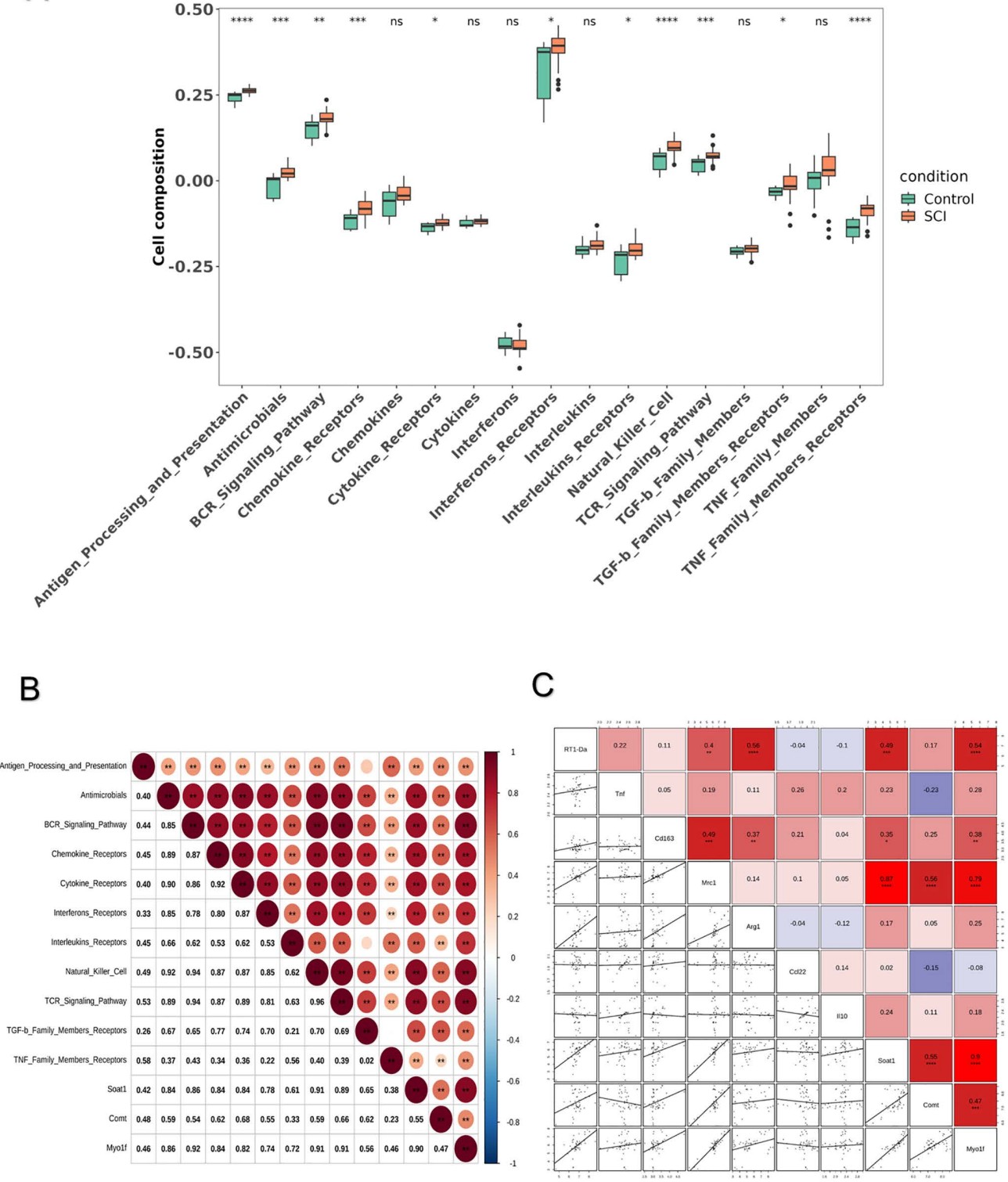

**Fig 6. Immune analysis between SCI and control groups. (A)** There were 11 immune response gene sets that were significantly differential between SCI and control groups. **(B)** Relevant analysis between hub genes and immune response. **(C)** The relevant analysis between hub genes and macrophage polarization markers.

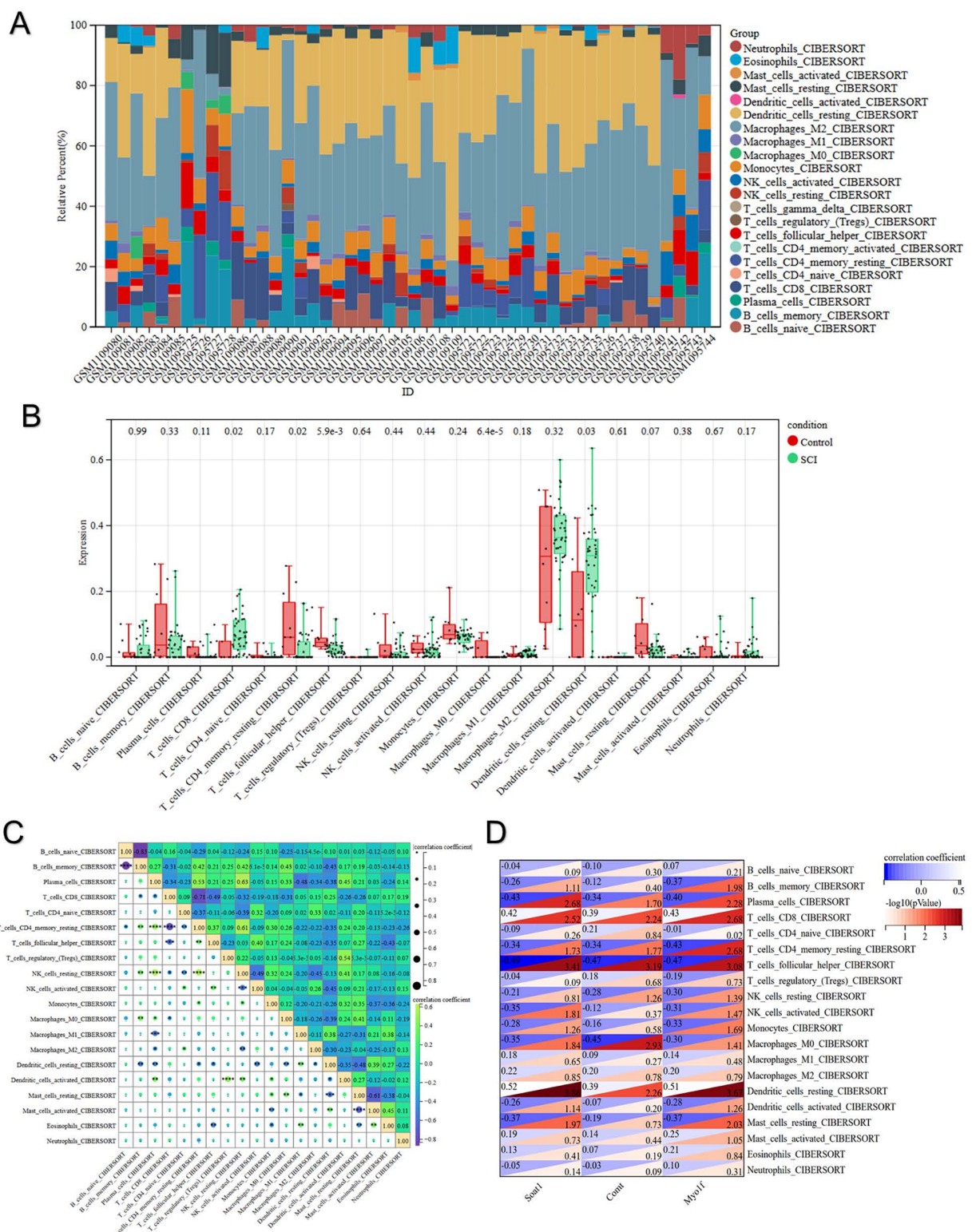

**Fig 7. Analysis of immune cell infiltration. (A)** The percentage stacked column plot illustrates the relative distribution of 22 immune cell types across each sample. **(B)** Box plots illustrate the levels of immune cell infiltration in the SCI and control groups. **(C)** The correlation heatmap illustrates the relationships between different immune cell compositions. **(D)** Heatmap of hub genes correlation with immune cells.

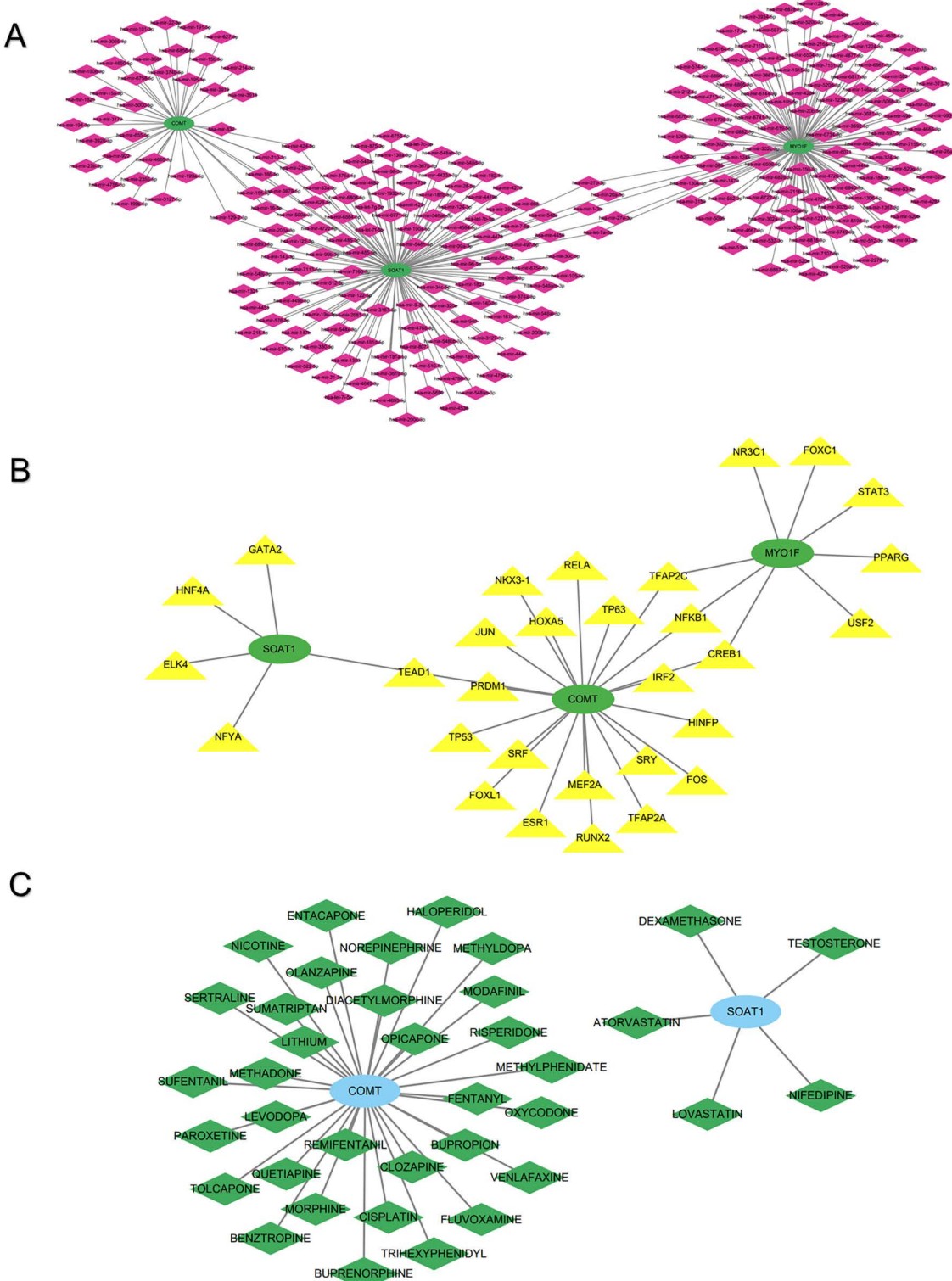

**Fig 8. The networks of hub genes. (A)** The miRNA-mRNA network of hub genes. **(B)** The TF-mRNA network of hub genes. **(C)** The hub gene-drug interaction network.

including fluvoxamine, tolcapone, and entacapone. However, no small-molecule drugs were predicted for *MYO1F* as the target gene.

### 3.9. Differential expression of hub genes between different macrophage polarization states

In GSE213240, a total of 66,093 cells and 19,404 genes met the inclusion criteria (S2A Fig, B). Subsequent analysis identified 2,000 highly variable genes with the greatest variance, such as *Igfbp5* (Fig 9A). Among these, the top 30 principal components (PCs) contributed the most to the variation (Fig 9B-9C). Clustering based on the top 30 PCs resulted in 17 cell clusters (Fig 9D). These clusters were classified into nine cell types, including T cells, B cells, M2 macrophages, M1 macrophages, oligodendrocytes, NK cells, endothelial cells, granulocytes, and monocytes. In SCI samples, granulocytes represented the largest proportion, followed by M1 macrophages, indicating an inflammatory microenvironment in SCI (Fig 9E). Notably, *Soat1*, *Comt*, and *Myo1f* showed significant differential expression between M1 and M2 macrophages, all being highly expressed in M1 macrophages (adjusted $P < 0.0001$, $\log_2FC > 1.50$) (Fig 9F). These findings further confirm the association of these hub genes with the inflammatory microenvironment of SCI.

### 3.10. The expression levels of the hub genes

The expression of hub genes between the SCI and control groups was further analyzed in the training set and the GSE183591 dataset using the Wilcoxon test (Fig 10A, 10B). The results demonstrated a significant difference in the expression of all hub genes between the SCI and control groups in the training set, with higher expression levels observed in the SCI group (Fig 10A). In the GSE183591 dataset, the expression trends of all hub genes were consistent with those in the training set (Fig 10B). These findings suggest that all hub genes exhibit strong diagnostic potential for SCI.

### 3.11. Validation of hub genes by RT-qPCR

RT-qPCR was performed on rat spinal cord tissue samples collected at 3, 7, and 14 days post-SCI to validate the reliability of the hub genes identified through database analysis. Consistent with the findings from the GEO databases, the expression levels of *Soat1*, *Comt*, and *Myo1f* were significantly elevated in the SCI group compared to the control group at all time points (Fig. 11, S6 Table). These results corroborated the data mining outcomes, further reinforcing the reliability of our findings.

## 4. Discussion

Activation of the innate immune response initiates post-SCI inflammatory cascades, with early stages chiefly defined by microglia and macrophage polarization [13,34]. Functional implications of DEGs underwent initial scrutiny in our work through GO and KEGG pathway enrichment evaluations. Substantial GO term accumulations for DEGs involved positive regulation of response to external stimuli, myeloid leukocyte activation, and immune response-regulating signaling pathway. KEGG pathway evaluations further disclosed DEG associations with pertussis pathway, NF-kappa B signaling pathway, and osteoclast differentiation. These results suggested that significant changes occur in the inflammatory microenvironment following SCI, and the associated MPRGs may play a crucial role in this process. LASSO, RF, and XGBoost algorithms were used to screen for hub genes; as a result, three hub genes (*Soat1*, *Comt*, and *Myo1f*) were identified. These results suggest that significant changes occur in the inflammatory microenvironment following SCI, and the associated MPRGs may play a critical role in this process. Sterol O-acyltransferase 1 (*SOAT1/ACAT1*) is a key enzyme in lipid metabolism. Recent studies have shown that inhibiting *SOAT1* promotes M2 macrophage polarization, reduces

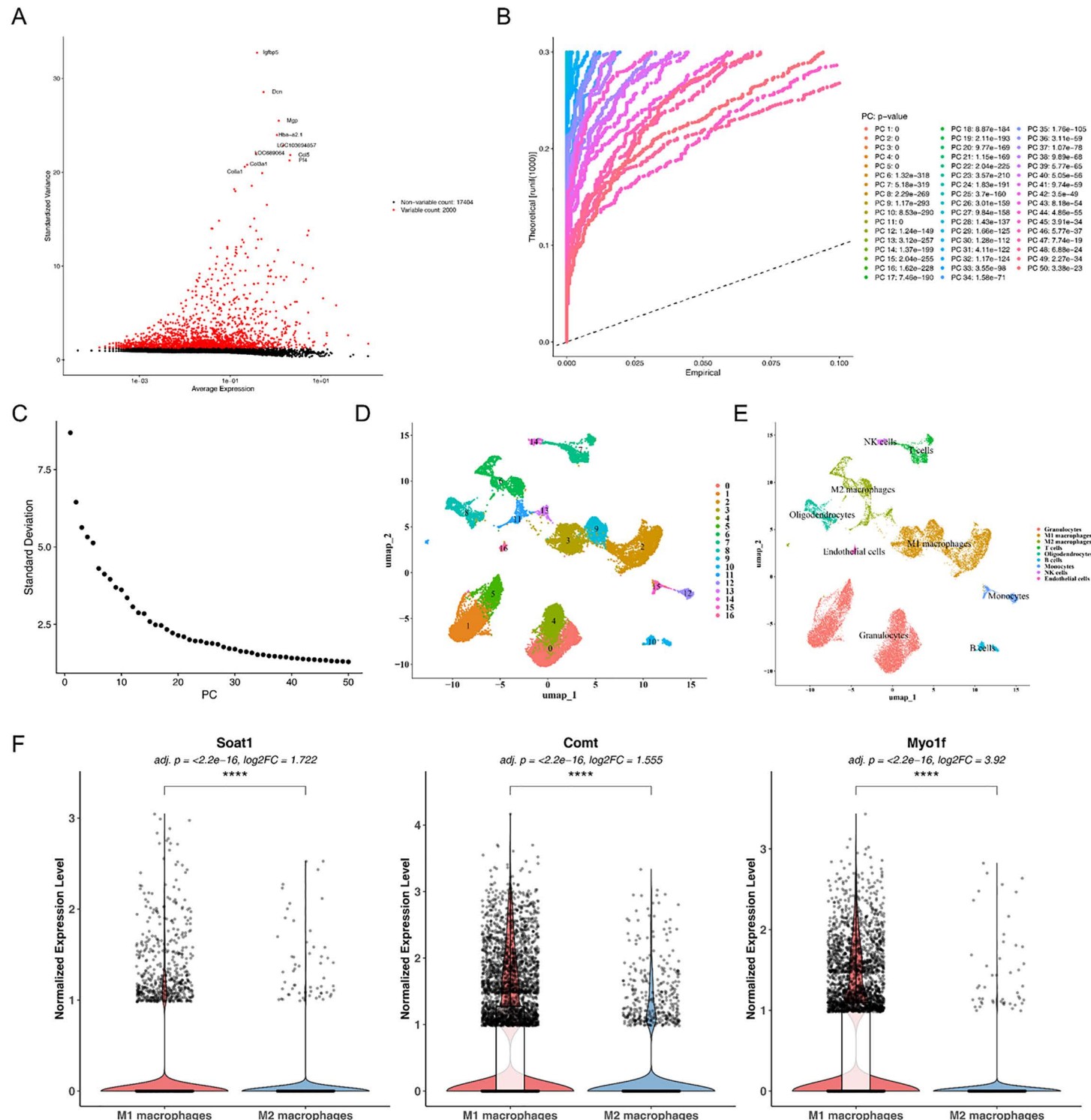

**Fig 9. Significant differences exist in the expression of hub genes between M1 and M2 macrophages. (A)** Visualization of highly variable genes. Red dots indicate highly variable genes, while black dots represent non-highly variable genes. **(B)** Jackstraw plot for PCA. The horizontal axis shows empirical values calculated by the Jackstraw function, the vertical axis displays computed theoretical values, and the legend on the right indicates the *P*-value for each principal component. **(C)** Number of available dimensions in the fragment plot. The horizontal axis represents the number of PCs, while

the vertical axis shows the standard deviation. **(D)** UMAP plot of cell clusters. The x-axis represents the first principal component axis after UMAP dimensionality reduction, while the y-axis represents the second principal component axis. Points of different colors denote distinct cell clusters. **(E)** UMAP plot of annotation results for 9 cell types across all samples. **(F)** Violin plots showing the expression levels of *Soat1*, *Comt*, and *Myo1f* in M1 and M2 macrophages.The overlaid box plot (white) illustrates the median and interquartile range, while the scatter plot displays the expression values of individual cells. Between-group comparisons were performed using the Wilcoxon rank-sum test with Bonferroni correction for multiple testing.

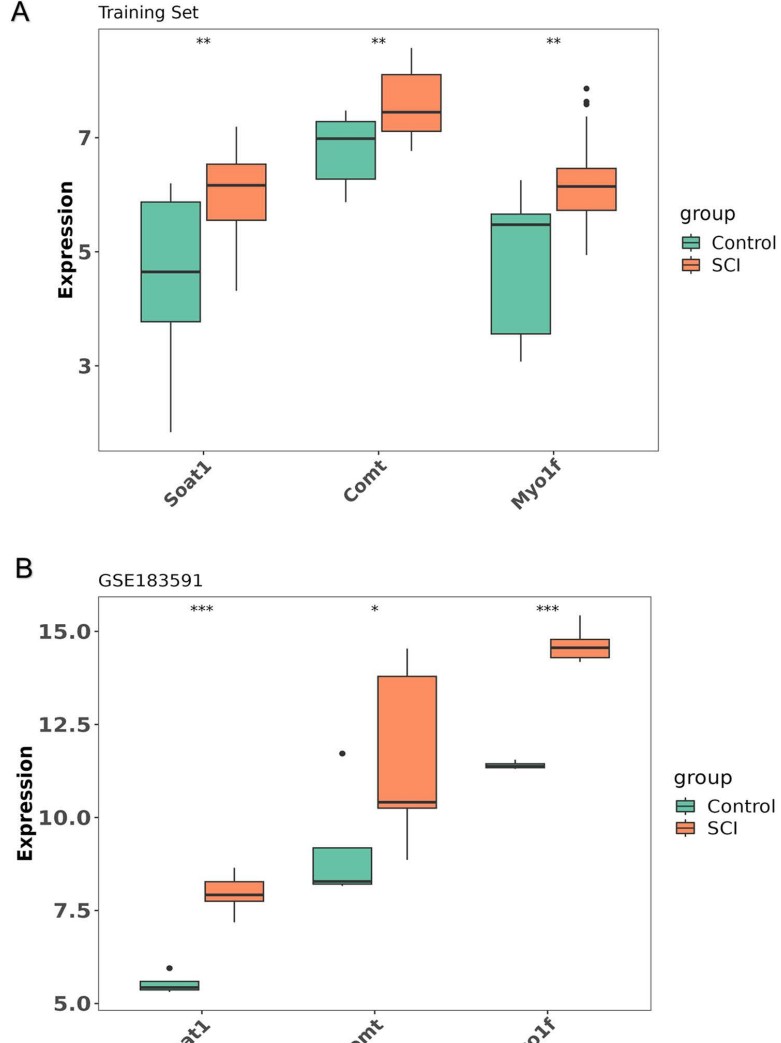

**Fig 10. The expression levels of the hub genes. (A)** The expression of all hub genes of SCI group was markedly higher in the training set. **(B)** In GSE183591, the expression of all hub genes of SCI group was markedly higher.

inflammatory responses, and supports functional recovery following SCI [35]. This finding suggests that modulating Soat1 activity may represent a promising therapeutic approach to enhance tissue repair after SCI. *Myo1f* (*Myosin IF*) encodes a non-conventional myosin that hydrolyzes ATP to generate mechanical force and supports actin-dependent cellular motility. Emerging evidence indicates that *Myo1f* is essential for neutrophil locomotion within three-dimensional matrices during

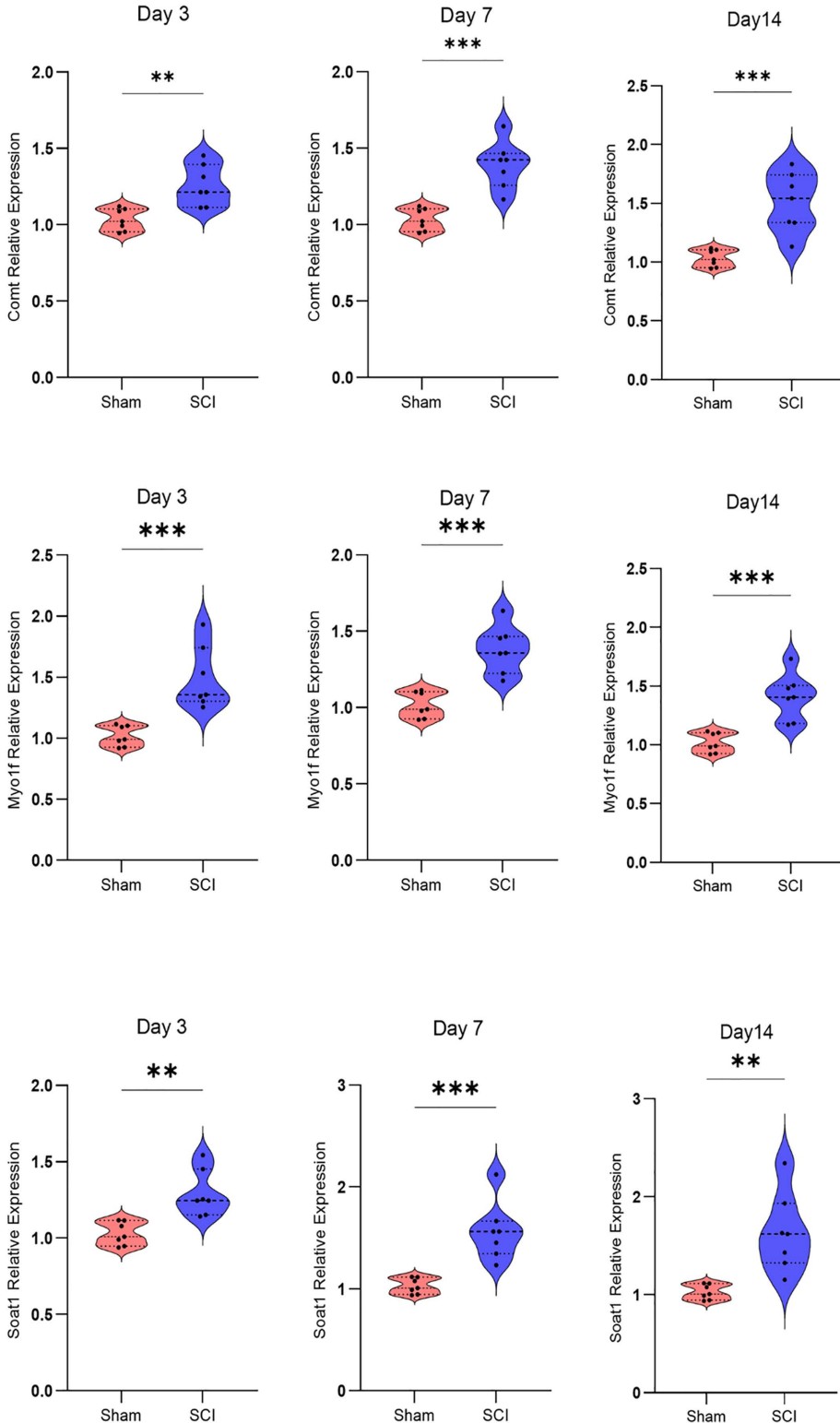

**Fig 11. Verification of differentially expressed hub gene mRNAs.** Relative mRNA expression levels of the *Comt*, *Myo1f*, and *Soat1* genes were assessed in both injured spinal cord and sham spinal cord samples by RT-qPCR at different time points. Data are expressed as the mean±SD (n=7). *P<0.05, **P<0.01, ***P<0.001 (Student's t-test).

acute inflammatory responses and is upregulated following SCI [36]. Furthermore, previous studies have demonstrated that genetic ablation of *Myo1f* attenuates microglial activation, dampens inflammatory signaling, and reduces overall immune reactivity [37]. In peripheral T-cell lymphoma, the Vav1–Myo1f fusion drives CD4 + T-cell transformation and disrupts normal T-cell differentiation, accompanied by increased tumor-associated macrophages (TAMs). In the corresponding mouse model, the spleen contains a distinct macrophage subset co-expressing M2 and TAM markers, suggesting a close mechanistic link between *Myo1f* and macrophage polarization [38]. *Myo1f* and *Soat1* regulate cell adhesion/migration and mediate cholesterol esterification metabolism, respectively. Both have been implicated in regulating macrophage polarization and directional chemotactic motility [39,40]. Catechol-O-methyltransferase (*Comt*) is an enzyme that facilitates O-methylation and effectively inactivates catechol-containing molecules. It has been reported to attenuate the anti-inflammatory effects of luteolin by promoting methylation [41]. *Comt* plays a central role in pain modulation, inflammatory regulation, and cognitive processes [42–44]. At the single-cell level, M1 macrophages represented the second-largest cell population, reflecting the pronounced inflammatory microenvironment at the injury site. The three hub genes exhibited distinct expression patterns between M1 and M2 polarization states, with consistent upregulation in M2 macrophages, indicating a close association with macrophage phenotypes and the local inflammatory milieu. These findings provide single-cell-level evidence linking the hub genes to macrophage functional states and the surrounding microenvironment, offering cellular insights into their potential role in immune modulation following central nervous system injury.

GSEA results indicated that all hub genes were involved in natural killer cell-mediated cytotoxicity, systemic lupus erythematosus, lysosome function, and complement and coagulation cascades. Moreover, immune-related gene set enrichment analysis revealed that *Soat1*, *Comt*, and *Myo1f* were significantly positively correlated with all 11 immune response gene sets that were differentially expressed between the SCI and control groups. Notably, NK cells play a crucial role in defending against bacterial and viral infections through their effector functions. They are also believed to significantly contribute to the immune dysfunction that arises following SCI. These findings suggest that the hub genes may have a pivotal role in SCI through various pathways, particularly those related to immune responses.

It is well established that miRNAs and TFs are essential regulators in the growth, development, and regeneration of the central nervous system (CNS). Given their regulatory roles in the pathophysiology of various diseases, we constructed miRNA-mRNA and TF-mRNA regulatory networks for the identified hub genes. Identifying the miRNAs and TFs within these networks may potentially shed light on the gene regulatory mechanisms governing the polarization of macrophages and microglia during SCI. For instance, it has been shown that inhibition of miR-27b-3p reduces microglial apoptosis after SCI and attenuates the production of inflammatory factors by microglia [45]. Another study demonstrated that miR-211-5p alleviates neuron apoptosis and inflammation induced by SCI by directly targeting activating transcription factor 6 (*ATF6*) and regulating endoplasmic reticulum stress in a rat mode [46]. The present analyses highlighted CREB1 and NFKB1 as key targets in the constructed TF-mRNA regulatory network, which are closely associated with *COMT* and *MYO1F*. A previous study demonstrated that pentraxin 3 (PTX3) promotes M2 macrophage polarization via the CREB1/CEBPB axis, thereby contributing to stromal cell-mediated immune regulation [47]. Another study revealed that NFKB1 is the most highly expressed TF in macrophages—key cellular drivers of inflammation and immunity [47,48]. The aforementioned findings may serve as important therapeutic targets for SCI and facilitate a deeper understanding of the mechanisms underlying the hub genes' roles in SCI.

We used the DGIdb to identify 37 small-molecule drugs predicted to interact with these hub genes. These agents will provide crucial guidance for further translational research and the development of viable therapeutic strategies for SCI. Interestingly, several of these drugs have already demonstrated efficacy in treating SCI in animal models and are associated with degenerative processes linked to SCI. For instance, nifedipine may help prevent adverse blood pressure responses after SCI. Evidence indicates that anti-inflammatory actions arise from its blockade of nitric oxide (NO) synthesis via inducible nitric oxide synthase (iNOS) within macrophages, coupled with interleukin-1β curtailment [49]. Functioning as a manufactured glucocorticoid, dexamethasone (Dex) curbs macrophage elaboration of pro-inflammatory

agents while curtailing apoptosis-associated cellular demise, in turn promoting expedited functional amelioration following SCI [50,51]. Another finding indicates that testosterone promotes macrophage polarization toward the M2 phenotype and suppresses M1 polarization through Gαi- and Akt-dependent signaling pathways [52]. Additionally, evidence supports that adjuvant testosterone, when used as part of a multimodal approach, facilitates neuromuscular recovery following SCI [52,53]. Atorvastatin and lovastatin have been reported to suppress the production of inflammatory mediators by macrophages and promote neurological function recovery in models of traumatic brain injury (TBI) and SCI [54–57]. In TBI models, fluvoxamine, a selective serotonin reuptake inhibitor (SSRI), has been shown to exert neuroprotective and anti-inflammatory effects [58]. However, further research is required to ascertain whether these drugs exert similar effects in SCI, as well as to investigate their underlying pharmacological mechanisms.

Despite providing novel bioinformatic insights into macrophage polarization in SCI, several limitations must be addressed before translating these findings into clinical applications. First, the reliance on bulk RNA sequencing limits the resolution to distinguish cell-subtype-specific gene expression, particularly among macrophage polarization states, thus restricting the precise functional interpretation of hub genes. While scRNA-seq analyses using tools such as Seurat enable cell clustering and differential expression analysis, they remain insufficient for accurately defining continuous polarization phenotypes or capturing low-abundance, polarization-specific transcripts, which may affect the interpretation of the macrophage polarization spectrum. Second, although hub genes such as *SOAT1*, *COMT*, and *MYO1F* were enriched in immune-related pathways, their direct roles in regulating macrophage phenotype transitions (e.g., M1/M2 balance), underlying mechanisms, and therapeutic potential remain largely unexplored, with current evidence limited to preliminary in vivo experiments. Third, the relatively small sample size may limit statistical power and introduce potential selection bias.

Future work should address these gaps in three areas: (1) increasing the sample size across diverse injury severities, disease stages, and demographic backgrounds to reduce bias and enhance generalizability; (2) employing macrophage-specific genetic manipulations, such as conditional knockouts or overexpression, to validate the direct effects on macrophage polarization, cytokine secretion, and functional recovery after SCI; and (3) applying spatial and co-localization techniques, such as double immunofluorescence, to verify hub gene expression in macrophages within tissue contexts, further validating these findings in clinical samples. Given the high disability and economic burden associated with SCI, validating these targets could provide molecular entry points for precise therapeutic interventions and establish potential biomarkers for dynamic monitoring and individualized treatment strategies in SCI and related neurological disorders.

## 5. Conclusions

In summary, macrophage polarization plays a critical role in the progression and functional outcomes of SCI. In this study, we comprehensively analyzed the expression profiles of macrophage polarization–related genes in SCI and, through systematic bioinformatics approaches, identified three hub genes (*Soat1*, *Comt*, and *Myo1f*) that are closely associated with this process. These findings deepen our understanding of the pathophysiological mechanisms underlying macrophage polarization–mediated neuroinflammation and tissue injury/repair following SCI, and provide novel therapeutic targets and a theoretical basis for clinical intervention. Based on our results, targeted modulation of *Soat1*, *Comt*, and *Myo1f* may represent a promising strategy to optimize macrophage polarization, suppress detrimental neuroinflammation, and ultimately maximize neuroprotection and functional recovery after SCI. Future studies should focus on gene-specific intervention strategies to facilitate the translation of these findings into effective therapeutic approaches.

## Supporting information

**S1 Table. The results of differential expression analysis between control and SCI samples in the training set.**
(XLSX)

**S2 Table. 1,297 GO items of DEGs.**
(XLSX)

**S3 Table. 126 enriched pathways of DEGs.**
(XLSX)

**S4 Table. The integrated results of the two datasets analyzed by RRA package.**
(XLSX)

**S5 Table. The enrichment results of gene set enrichment analysis (GSEA) for the hub genes.**
(XLSX)

**S6 Table. RT-qPCR data for selected hub genes.**
(XLSX)

**S1 Fig. Batch effects were removed after merging the GSE45550 and GSE45006 datasets.** (A) Before batch effect correction. (B) After batch effect correction, showing excellent batch integration.
(TIF)

**S2 Fig. The quality control of single-cell RNA sequencing (scRNA-seq) data.** (A, B) Distribution plots of nFeature_RNA, nCount_RNA, and percent.mt before (A) and after quality control (B).
(TIF)

**S1 File. ARRIVE Checklist.**
(PDF)

**S2 File. Inclusivity in global research questionnaire.**
(DOCX)

**S3 File. PLOS ONE humane endpoints checklist.**
(DOCX)

## Author contributions

**Conceptualization:** Xiaowei Zha, Shen Cao.

**Formal analysis:** Xiaowei Zha.

**Funding acquisition:** Shen Cao.

**Project administration:** Shen Cao.

**Resources:** Shen Cao.

**Software:** Xiaowei Zha.

**Supervision:** Shen Cao.

**Writing – original draft:** Xiaowei Zha.

**Writing – review & editing:** Xiaowei Zha, Shen Cao.

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
