## [Decision Letter · Decision Letter 0]

10 Oct 2025

PONE-D-25-22705Screening macrophage polarization genes in spinal cord injury as therapeutic targetsPLOS ONE

Dear Dr. Cao,

Thank you for submitting your manuscript to PLOS ONE. After careful consideration, we feel that it has merit but does not fully meet PLOS ONE’s publication criteria as it currently stands. Therefore, we invite you to submit a revised version of the manuscript that addresses the points raised during the review process.

If applicable, we recommend that you deposit your laboratory protocols in protocols.io to enhance the reproducibility of your results. Protocols.io assigns your protocol its own identifier (DOI) so that it can be cited independently in the future. For instructions see: https://journals.plos.org/plosone/s/submission-guidelines#loc-laboratory-protocols. Additionally, PLOS ONE offers an option for publishing peer-reviewed Lab Protocol articles, which describe protocols hosted on protocols.io. Read more information on sharing protocols at . Additionally, PLOS ONE offers an option for publishing peer-reviewed Lab Protocol articles, which describe protocols hosted on protocols.io. Read more information on sharing protocols at https://plos.org/protocols?utm_medium=editorial-email&utm_source=authorletters&utm_campaign=protocols..

We look forward to receiving your revised manuscript.

Kind regards,

Sawar Khan, Ph.D

Academic Editor

PLOS ONE

Journal Requirements:

https://www.sciencedirect.com/science/article/pii/S2950588725000424?via%3Dihub

In your revision ensure you cite all your sources (including your own works), and quote or rephrase any duplicated text outside the methods section. Further consideration is dependent on these concerns being addressed.

“This work was supported by General Programs of Natural Science Research of Anhui Provincial Education Department (Grant No. ZR2022B001).”

Reviewers' comments:

Reviewer's Responses to Questions

**Comments to the Author**

1. Is the manuscript technically sound, and do the data support the conclusions?

Reviewer #1: Yes

Reviewer #2: Yes

2. Has the statistical analysis been performed appropriately and rigorously? 

Reviewer #1: Yes

Reviewer #2: Yes

3. Have the authors made all data underlying the findings in their manuscript fully available?

Reviewer #1: No

Reviewer #2: No

4. Is the manuscript presented in an intelligible fashion and written in standard English?

Reviewer #1: No

Reviewer #2: Yes

5. Review Comments to the Author

Reviewer #1: Introductory comment

In the manuscript titled “Screening macrophage polarization genes in spinal cord injury as therapeutic targets”, Xiaowei et al. attempt to explore an understudied area of macrophage polarization in spinal cord injury (SCI). To achieve this, the authors adopted a fairly comprehensive bioinformatics analysis, ranging from screening differential gene expression analysis (MPRGs) to miRNA/TF regulatory network construction. Furthermore, the authors explored external validation of the three final DEGs (via qPCR testing). Overall, the study is sound, timely, and worth considering. However, the manuscript can further be fine-tuned by addressing a couple of sections.

Core strengths

-----------------

Standard bioinformatics pipeline: Robust feature-selection techniques (random forest, LASSO, and gradient boosting) and enrichment methods were used to screen the differentially expressed genes. Performing batch correction via the sva package is particularly commendable.

Immunoinfiltration and function pathway analysis: The authors used ssGSEA to highlight functional pathway enrichment of the screened genes, as well as using CIBERSORT to underscore the role of the identified genes in the immune cell infiltration within the tumor microenvironment.

Regulatory and gene-drug interaction network: The authors further performed TF-miRNA and drug-gene interaction analysis to illustrate or reveal translational relevance.

External validation: Though not a strong point, performing in vivo qPCR validation in rat models adds further insights (with respect to predictions).

Weaknesses

--------------

Pre-processing: Though it is mentioned that batch correction was performed (i.e., using the sva package), the manuscript is quite silent on key data pre-processing methods. This aspect should be addressed. The datasets used are described in moderate detail (e.g., platform types like GPL1355 and GPL4135). Additionally, the sample sizes are small, which implies the authors ought to describe how they dealt with sample-size bias.

Lack of in-depth discussion of the SCI hub genes: A detailed contextual discussion of the screened genes in SCI would further strengthen the manuscript. While gene ontology (GO/KEGG) was performed, discussion of the MPGRs with respect to past studies has not been thoroughly discussed.

Language clarity: The language can further be improved. There are some weird spelling errors (aftereffects → after-effects). Italicize the ‘genus’ and ‘species’ names (e.g., Rattus norvegicus → Rattus norvegicus). Using the word ‘gained’ in the expression “Firstly, the DE-MPRGs were gained through taking the intersection” does not quite appear standard.

Limitations of the study: Clearly, this study has a couple of limitations not mentioned in the manuscript. For example, single-cell RNA analysis (Seurat package) was performed. Therefore, the authors could dedicate a paragraph to pointing out areas lacking in the current study, as well as further direction (if any).

Reviewer #2: The manuscript is comprehensive and methodologically sound, providing a solid foundation for the study’s conclusions. The study offers a comprehensive bioinformatics analysis aimed at identifying macrophage polarization-related genes involved in spinal cord injury (SCI). The multi-step approach —encompassing DEG screening, GO/KEGG enrichment analyses, machine learning algorithms (LASSO, XGBoost, RF), gene-gene interaction networks, and drug prediction — is well-designed and leads to valuable insights. Addressing the following points with additional details and clarifications would improve transparency, and reproducibility.

- It would enhance reproducibility if the authors provide more information on the specific parameters or methods used for batch effect correction. Additionally, including a visualization (e.g., PCA or MDS plots) before and after batch effect removal would help readers assess the effectiveness of this step.

- I recommend using GridSearchCV to improve the performance of each machine learning algorithms.

- The application of the “limma” package with clear thresholds (P < 0.05, |log2FC| > 0.5) is appropriate for identifying DEGs. Functional enrichment via GO and KEGG pathways using “clusterProfiler” and “org.Rn.eg.db” packages is well described. Nevertheless, the authors should clarify whether multiple testing corrections (e.g., FDR adjustment) were applied and specify the exact adjusted P-value thresholds for enrichment analyses.

- Employing three machine learning algorithms (LASSO, Random Forest, and XGBoost) for biomarker selection is a strong approach. However, detailed information about the tuning parameters, cross-validation strategy, and feature selection criteria for each algorithm is missing and should be provided to enhance reproducibility. Also, a brief description of how the final biomarker set was validated (e.g., internal validation metrics or external validation) would strengthen this section.

- Given the study's focus on macrophage polarization, incorporating or referencing single-cell transcriptomics data would improve resolution and specificity. The authors are encouraged to discuss this limitation more explicitly in the discussion section

- To firmly establish the involvement of the identified hub genes in macrophages, the use of co-localization techniques (e.g., double immunofluorescence or in situ hybridization) is recommended. This would strengthen the claim that these genes are functionally active in macrophage populations post-SCI.

- Although the study briefly discusses the biological roles of *Soat1*, *Comt*, and *Myo1f*, a more in-depth analysis of their functions in neuroinflammation and SCI — supported by relevant literature — would be valuable.

- Literatures are taken from old papers where enhanced version of clustering techniques are available. So, add some more recent literatures and compare the work with proposed one.

- The section about the future study should be mentioned in the last paragraph of the Conclusion section.

- Please expand on how the findings can be practically integrated into clinical or public health strategies?

- The quality of the figures should be improved.

6. PLOS authors have the option to publish the peer review history of their article (what does this mean?). If published, this will include your full peer review and any attached files.). If published, this will include your full peer review and any attached files.

.

Reviewer #1: **Yes:** Abubakari Sumaila SalpawuniAbubakari Sumaila Salpawuni

Reviewer #2: No

While revising your submission, please upload your figure files to the Preflight Analysis and Conversion Engine (PACE) digital diagnostic tool, https://pacev2.apexcovantage.com/. PACE helps ensure that figures meet PLOS requirements. To use PACE, you must first register as a user. Registration is free. Then, login and navigate to the UPLOAD tab, where you will find detailed instructions on how to use the tool. If you encounter any issues or have any questions when using PACE, please email PLOS at . PACE helps ensure that figures meet PLOS requirements. To use PACE, you must first register as a user. Registration is free. Then, login and navigate to the UPLOAD tab, where you will find detailed instructions on how to use the tool. If you encounter any issues or have any questions when using PACE, please email PLOS at figures@plos.org. Please note that Supporting Information files do not need this step.. Please note that Supporting Information files do not need this step.

---

## [Author Response · Author response to Decision Letter 1]

4 Dec 2025

Journal Requirements:

Re: We sincerely thank the PLOS ONE editorial team for reviewing our manuscript and providing valuable comments. We have carefully considered all suggestions and have thoroughly revised the manuscript to comply with the formatting requirements and editorial guidance provided.

Re: We have carefully reviewed PLOS’ policy on inclusivity in global research and completed the corresponding questionnaire as required, and appropriately cited it in the Materials and methods section (Line 280-282). The completed questionnaire has been included as a Supporting Information file with the revised manuscript, and we believe this addition further enhances the transparency of our study.

https://www.sciencedirect.com/science/article/pii/S2950588725000424?via%3Dihub

In your revision ensure you cite all your sources (including your own works), and quote or rephrase any duplicated text outside the methods section. Further consideration is dependent on these concerns being addressed.

Re: We sincerely thank the editor for the careful review of our manuscript and for providing valuable comments regarding textual presentation. We have taken the issue of textual overlap very seriously and conducted a thorough check and revision of the entire manuscript. During the revision process, we carefully cross-checked all cited references and substantively rewrote and restructured sections of the Introduction, Discussion, and other parts where overlaps were identified, ensuring originality in language expression. While some bioinformatic analysis methods, such as hub gene selection via machine learning, are standard in the field, the novelty of this study lies in the integration of transcriptomic data with single-cell RNA-seq, allowing higher-resolution analysis of macrophage subpopulation dynamics, and in the experimental validation of selected hub genes in vivo—work that has not been systematically reported in the existing literature. We have conducted a similarity/plagiarism check on the manuscript and confirm that the revised version fully addresses any issues of textual overlap, ensuring original language and proper citation throughout. We appreciate your guidance, which has helped us more clearly highlight the unique contributions of this study, and we respectfully submit the manuscript for further consideration.

“This work was supported by General Programs of Natural Science Research of Anhui Provincial Education Department (Grant No. ZR2022B001).”

Re: Thank you very much for reviewing our manuscript and providing valuable comments. In response to the request for an updated financial disclosure statement, we have complied accordingly. The funder (General Programs of Natural Science Research of Anhui Provincial Education Department, Grant No. ZR2022B001) had no role in any aspect of this study. Accordingly, we have included the following revised statement in our cover letter:

"This work was supported by the General Programs of Natural Science Research of Anhui Provincial Education Department (Grant No. ZR2022B001). The funders had no role in study design, data collection and analysis, decision to publish, or preparation of the manuscript."

We confirm that this statement is accurate. We would appreciate the journal’s assistance in updating the corresponding information in the online submission system. Thank you again for your guidance.

Re: We sincerely thank the editor for reviewing our manuscript and for providing valuable comments. We have taken note of the suggestion regarding reference citations. We carefully reviewed and evaluated the publications mentioned by the reviewers and have cited them judiciously in the revised manuscript where appropriate.

Reviewers' comments:

Reviewer's Responses to Questions

Comments to the Author

1. Is the manuscript technically sound, and do the data support the conclusions?

Reviewer #1: Yes

Reviewer #2: Yes

Re: We sincerely thank you for your recognition of the technical rigor, data reliability, and validity of the conclusions in our study. We are pleased that the scientific quality of our manuscript has been acknowledged. We remain committed to rigorous research and have carefully revised and improved the manuscript based on the other suggestions provided.

2. Has the statistical analysis been performed appropriately and rigorously?

Reviewer #1: Yes

Reviewer #2: Yes

Re: We sincerely thank you for acknowledging the rigor and appropriateness of the statistical analyses in our study. We remain committed to conducting our research with objective and rigorous statistical standards.

3. Have the authors made all data underlying the findings in their manuscript fully available?

Reviewer #1: No

Reviewer #2: No

Re: Thank you for reviewing our manuscript and providing valuable comments. We fully recognize the shortcomings regarding data availability and take this issue seriously. To comply with PLOS data policies, we have submitted the raw data underlying all statistical figures/charts as Supporting Information, and updated the Data Availability Statement to include repository names, accession numbers, and direct access links, ensuring that all data are publicly accessible without restriction. Additionally, key datasets supporting critical results are provided alongside the manuscript to enhance reproducibility and transparency. We are committed to fully implementing these changes in the revised manuscript, ensuring that all data are completely accessible, traceable, and in compliance with journal requirements, and we sincerely appreciate your suggestions for improving the scientific rigor of our work.

4. Is the manuscript presented in an intelligible fashion and written in standard English?

Reviewer #1: No

Reviewer #2: Yes

Re: We sincerely thank the reviewer for the valuable comments, particularly regarding the clarity and precision of the language. In response to the issues highlighted by Reviewer #1, we have thoroughly polished the manuscript, correcting grammatical and spelling errors, inappropriate expressions, and potentially ambiguous sentences, while also optimizing the logic and structure and reorganizing some lengthy or obscure paragraphs to ensure clear, accurate, and easily understandable presentation. The revised manuscript reflects all these improvements, and we greatly appreciate the meticulous work of the reviewers and editors, which has been instrumental in enhancing the quality of our study.

5. Review Comments to the Author

Reviewer #1: Introductory comment

In the manuscript titled “Screening macrophage polarization genes in spinal cord injury as therapeutic targets”, Xiaowei et al. attempt to explore an understudied area of macrophage polarization in spinal cord injury (SCI). To achieve this, the authors adopted a fairly comprehensive bioinformatics analysis, ranging from screening differential gene expression analysis (MPRGs) to miRNA/TF regulatory network construction. Furthermore, the authors explored external validation of the three final DEGs (via qPCR testing). Overall, the study is sound, timely, and worth considering. However, the manuscript can further be fine-tuned by addressing a couple of sections.

Core strengths

-----------------

Standard bioinformatics pipeline: Robust feature-selection techniques (random forest, LASSO, and gradient boosting) and enrichment methods were used to screen the differentially expressed genes. Performing batch correction via the sva package is particularly commendable.

Immunoinfiltration and function pathway analysis: The authors used ssGSEA to highlight functional pathway enrichment of the screened genes, as well as using CIBERSORT to underscore the role of the identified genes in the immune cell infiltration within the tumor microenvironment.

Regulatory and gene-drug interaction network: The authors further performed TF-miRNA and drug-gene interaction analysis to illustrate or reveal translational relevance.

External validation: Though not a strong point, performing in vivo qPCR validation in rat models adds further insights (with respect to predictions).

Weaknesses

--------------

Pre-processing: Though it is mentioned that batch correction was performed (i.e., using the sva package), the manuscript is quite silent on key data pre-processing methods. This aspect should be addressed. The datasets used are described in moderate detail (e.g., platform types like GPL1355 and GPL4135). Additionally, the sample sizes are small, which implies the authors ought to describe how they dealt with sample-size bias.

Re: We sincerely thank you for your recognition of the value of our study and for your insightful comments regarding data preprocessing and sample size, which have been crucial in enhancing the rigor and clarity of our manuscript. In response, we have expanded the “Materials and Methods” section to provide detailed information on the quality control steps performed prior to batch correction and to explain the statistical rationale for the chosen preprocessing procedures, ensuring data reliability and comparability. We have also addressed concerns regarding the limited sample size by emphasizing in the Methods the use of robust statistical approaches and cross-validation techniques to reduce the risk of overfitting, and by acknowledging in the Discussion this limitation while highlighting the need for validation in larger independent cohorts. These revisions aim to present the study’s limitations more comprehensively and demonstrate our careful consideration of these issues. We greatly appreciate your insightful comments, which have significantly improved the quality of our manuscript, and we hope that these modifications fully address your concerns.

Lack of in-depth discussion of the SCI hub genes: A detailed contextual discussion of the screened genes in SCI would further strengthen the manuscript. While gene ontology (GO/KEGG) was performed, discussion of the MPGRs with respect to past studies has not been thoroughly discussed.

Re: We sincerely thank you for this insightful and constructive comment. We fully agree that a thorough comparison and interpretation of the selected hub genes within the context of existing literature is crucial for enhancing the academic value of the manuscript. Following your suggestion, we have revised and expanded the Discussion section. Rather than merely describing the results of GO/KEGG enrichment analyses, we have examined each of the final hub genes in the context of current literature, deepening the discussion of their known functions in SCI or other neurological disorders and the consistency of our findings with previous studies. We have also integrated insights from the literature and single-cell analyses to elucidate the relationships between these genes and macrophage polarization, highlighting the novelty of our study in identifying potential key targets. We believe that these additions and elaborations now allow the Discussion section to more fully interpret the potential significance of our findings. We greatly appreciate your guidance, which has significantly strengthened the depth of our manuscript.

Language clarity: The language can further be improved. There are some weird spelling errors (aftereffects → after-effects). Italicize the ‘genus’ and ‘species’ names (e.g., Rattus norvegicus → Rattus norvegicus). Using the word ‘gained’ in the expression “Firstly, the DE-MPRGs were gained through taking the intersection” does not quite appear standard.

Limitations of the study: Clearly, this study has a couple of limitations not mentioned in the manuscript. For example, single-cell RNA analysis (Seurat package) was performed. Therefore, the authors could dedicate a paragraph to pointing out areas lacking in the current study, as well as further direction (if any).

Re: We sincerely thank you for your specific and valuable suggestions regarding language details and the limitations of our study. We have carefully revised the manuscript based on each of your comments.

1. Language clarity: We have thoroughly proofread and polished the entire manuscript. Specific revisions include correcting spelling errors, ensuring proper formatting of all biological genus and species names, and replacing non-standard expressions with more precise academic language.

2. Study limitations: We fully agree with your points and have added a dedicated section on limitations and future directions at the end of the Discussion. This section addresses potential biases associated with retrospective analyses based on public databases, the inherent limitations of bulk RNA-seq data in distinguishing cell types, the current shortcomings of scRNA-seq analyses in precisely defining continuous phenotypes, and the preliminary nature of the

---

## [Decision Letter · Decision Letter 1]

18 Feb 2026

PONE-D-25-22705R1Screening macrophage polarization genes in spinal cord injury as therapeutic targetsPLOS One

Dear Dr. Cao,

Thank you for submitting your manuscript to PLOS ONE. After careful consideration, we feel that it has merit but does not fully meet PLOS ONE’s publication criteria as it currently stands. Therefore, we invite you to submit a revised version of the manuscript that addresses the points raised during the review process.

If applicable, we recommend that you deposit your laboratory protocols in protocols.io to enhance the reproducibility of your results. Protocols.io assigns your protocol its own identifier (DOI) so that it can be cited independently in the future. For instructions see: https://journals.plos.org/plosone/s/submission-guidelines#loc-laboratory-protocols. Additionally, PLOS ONE offers an option for publishing peer-reviewed Lab Protocol articles, which describe protocols hosted on protocols.io. Read more information on sharing protocols at . Additionally, PLOS ONE offers an option for publishing peer-reviewed Lab Protocol articles, which describe protocols hosted on protocols.io. Read more information on sharing protocols at https://plos.org/protocols?utm_medium=editorial-email&utm_source=authorletters&utm_campaign=protocols..

We look forward to receiving your revised manuscript.

Kind regards,

Sawar Khan, Ph.D

Academic Editor

PLOS One

Journal Requirements:

Reviewer's Responses to Questions

**Comments to the Author**

1. If the authors have adequately addressed your comments raised in a previous round of review and you feel that this manuscript is now acceptable for publication, you may indicate that here to bypass the “Comments to the Author” section, enter your conflict of interest statement in the “Confidential to Editor” section, and submit your "Accept" recommendation.

Reviewer #1: All comments have been addressed

Reviewer #3: (No Response)

2. Is the manuscript technically sound, and do the data support the conclusions?

Reviewer #1: Yes

Reviewer #3: Yes

3. Has the statistical analysis been performed appropriately and rigorously? 

Reviewer #1: Yes

Reviewer #3: Yes

4. Have the authors made all data underlying the findings in their manuscript fully available?

Reviewer #1: Yes

Reviewer #3: Yes

5. Is the manuscript presented in an intelligible fashion and written in standard English?

Reviewer #1: Yes

Reviewer #3: Yes

6. Review Comments to the Author

Reviewer #1: This revised manuscript has been significantly improved. In the original version, my concerns largely revolved around issues of preprocessing (e.g., small sample size), lack of detailed discussion of the SCI hub genes screened, language clarity, and failure to acknowledge the limitations of the present study. All these issues have largely been addressed now.

Minor suggestion:

Since the GSE45006 and GSE45550 datasets were used to screen the DEGs (using the `limma` package) for the downstream analyses, the findings in the study can further be strengthened using the DExMA package or the `Robust Rank Aggregation` (RRA) package for meta-analysis of the GSE45006 and GSE45550 datasets. Using these two packages, can the approach used in DOI: https://doi.org/10.3390/diagnostics15141770 be applicable to your study? How are your findings strengthened?

Since links to web tools are susceptible to changes in the future, the authors are strongly advised to include the date accessed for the web links in their study. As an example, the authors state

“Queries in the Drug-Gene Interaction Database (DGIdb; https://dgidb.genome.wustl.edu/) relied on individual hub genes …”

It should be

“Queries in the Drug-Gene Interaction Database (DGIdb; https://dgidb.genome.wustl.edu/ [Accessed: June 23, 2025]) relied on individual hub genes …” Notice that in the second case, we indicate the web tool was accessed on June 23, 2025.

Reviewer #3: I reviewed the revised manuscript which respond to the comments of previous reviewers who mostly focused on the data gathering and bioinformatic analysis. In summary this manuscript tried to identify the genes involved in macrophages polarization after SCI by recruiting data bases and doing bioinformatic analyses. They find three genes in their research and then bring them to the bench in order to evaluate their mRNA expression. I have some comments to more polish the animal study.

- Duration of receiving pain killers and antibiotic should be stated.

- clear group description and number of animals in each group required.

- use one format of genes names. International abbreviation usage is required such as Comt change to COMT, Myo1f to MYO1F and so on

- fig-9- To me the level of expression of desired gene in M1 and M2 macrophages does not have significant difference.

- for functionality of a gene usually we check the end product of a gene which is protein level. I didn't see any western blot or ELISA study in this work.

- I need to see a clear suggestion for future studies in conclusion section showing us based on this study who can we reach to maximum protection in SCI by manipulating these genes.

- Taken together, this study brings new hypothesis which needs to be evaluated by further animal and clinical trial studies.

7. PLOS authors have the option to publish the peer review history of their article (what does this mean?). If published, this will include your full peer review and any attached files.). If published, this will include your full peer review and any attached files.

.

Reviewer #1: No

Reviewer #3: **Yes:** Mahmoudreza HadjighassemMahmoudreza Hadjighassem

---

## [Author Response · Author response to Decision Letter 2]

24 Mar 2026

Point-by-point response to the editor and reviewer comments

Dear Editor and Reviewers,

Thank you very much for your continued evaluation of our manuscript and for your insightful comments and valuable suggestions. We sincerely appreciate your time and effort throughout the review process.

We have carefully revised the manuscript again in response to all the comments raised in this round of review. A detailed, point-by-point response to each comment is provided below. All changes have been clearly indicated in the revised manu-script. We have made every effort to address all remaining concerns and further im-prove the clarity, rigor, and completeness of the study.

We hope that the revised manuscript now meets the requirements for publication in PLOS ONE. Please do not hesitate to contact us if further clarification is needed.

Journal Requirements:

Re: N/A。

2. Please review your reference list to ensure that it is complete and correct. If you have cited papers that have been retracted, please include the rationale for doing so in the manuscript text, or remove these references and replace them with relevant current references. Any changes to the reference list should be mentioned in the re-buttal letter that accompanies your revised manuscript. If you need to cite a retracted article, indicate the article’s retracted status in the References list and also include a citation and full reference for the retraction notice.

Re: Thank you for your suggestion. We have carefully reviewed and standardized the reference list to ensure its accuracy and completeness. We identified that Reference 35 in the previous version, Retracted Article: Strontium-doped gelatin scaffolds promote M2 macrophage switch and angiogenesis through modulating the polariza-tion of neutrophils, had been retracted.

In accordance with the journal’s requirements, we removed this reference and re-placed it with a relevant and valid article of similar scientific significance: Macro-phage polarization-related gene SOAT1 is involved in inflammatory response and functional recovery after spinal cord injury. The manuscript has been updated ac-cordingly.

Review Comments to the Author

Reviewer #1: This revised manuscript has been significantly improved. In the origi-nal version, my concerns largely revolved around issues of preprocessing (e.g., small sample size), lack of detailed discussion of the SCI hub genes screened, language clarity, and failure to acknowledge the limitations of the present study. All these is-sues have largely been addressed now.

Re：Thank you very much for your careful evaluation and valuable comments. We are pleased that the current revision has largely addressed the concerns you previ-ously raised, including data preprocessing, in-depth discussion of SCI-related hub genes, language clarity, and the acknowledgment of study limitations. We sincerely appreciate your professional guidance.

Minor suggestion:

Since the GSE45006 and GSE45550 datasets were used to screen the DEGs (using the `limma` package) for the downstream analyses, the findings in the study can fur-ther be strengthened using the DExMA package or the `Robust Rank Aggregation` (RRA) package for meta-analysis of the GSE45006 and GSE45550 datasets. Using these two packages, can the approach used in DOI: https://doi.org/10.3390/diagnostics15141770 be applicable to your study? How are your findings strengthened?

Re：We sincerely thank you for your valuable suggestion. In response, we have per-formed additional analyses, and the results have been incorporated into the revised manuscript (without track changes) (lines 151–157, 350-353). The corresponding results have also been provided in the Supplementary Materials for your review.

Specifically, after identifying the hub genes, we applied the same significance thresholds to screen differentially expressed genes (DEGs) from the GSE45006 and GSE45550 datasets. We then used the RobustRankAggreg (RRA) package to inte-grate the ranked DEG lists from the two datasets. The aggregateRanks function was employed to calculate integrated scores and generate a cross-dataset ranked gene list. Based on this, we evaluated the ranking positions of the three hub genes (Soat1, Myo1f, and Comt) within the integrated list.

The rankings and corresponding RRA scores of the three hub genes are as follows: Soat1 ranked 1,336 (top 8.84%, Score = 0.0936), Myo1f ranked 1,605 (top 10.62%, Score = 0.1139), and Comt ranked 2,972 (top 19.66%, Score = 0.2258). Although these genes were not among the top 20 or top 100 in the integrated list, they were consistently retained in the RRA results and ranked within the upper-middle range across all genes.

Since links to web tools are susceptible to changes in the future, the authors are strongly advised to include the date accessed for the web links in their study. As an example, the authors state“Queries in the Drug-Gene Interaction Database (DGIdb; https://dgidb.genome.wustl.edu/) relied on individual hub genes …”

It should be

“Queries in the Drug-Gene Interaction Database (DGIdb; https://dgidb.genome.wustl.edu/ [Accessed: June 23, 2025]) relied on individual hub genes …” Notice that in the second case, we indicate the web tool was accessed on June 23, 2025.

Re：Thank you very much for your valuable suggestion. In response, we have added the specific access dates for all web-based tools used in this study and standardized the citation format of online resources. These revisions have been incorporated into the manuscript, improving the rigor and clarity of our presentation.

Reviewer #3: I reviewed the revised manuscript which respond to the comments of previous reviewers who mostly focused on the data gathering and bioinformatic analysis. In summary this manuscript tried to identify the genes involved in macro-phages polarization after SCI by recruiting data bases and doing bioinformatic anal-yses. They find three genes in their research and then bring them to the bench in or-der to evaluate their mRNA expression. I have some comments to more polish the animal study.

-Duration of receiving pain killers and antibiotic should be stated.

-clear group description and number of animals in each group required.

Re：We sincerely thank you for your valuable suggestions, which have greatly con-tributed to improving the methodological clarity and rigor of the animal experiments in this study. In response to your comments regarding the need for a clear descrip-tion of experimental groups and the number of animals in each group, as well as the duration of analgesic and antibiotic administration, we have carefully revised and supplemented the Methods section as follows:

At the beginning of the animal model description, we have clearly defined the exper-imental groups as one sham-operated group and three SCI groups corresponding to 3, 7, and 14 days post-injury. We have explicitly stated the initial number of animals in each group (8 rats per SCI group and 7 rats in the sham group), as well as the final number of animals included in the analyses (7 rats per group across all four groups).

In addition, in the postoperative care section, all analgesics and antibiotics were ad-ministered daily after surgery and continued for up to seven consecutive days, or un-til the day of sacrifice for animals in the 3-day group. Treatment was discontinued once rats regained spontaneous urination and showed no obvious signs of pain or in-fection.

All relevant revisions have been clearly indicated in the revised manuscript (see Section 2.12, “Spinal cord injury rat model”).

- use one format of genes names. International abbreviation usage is required such as Comt change to COMT, Myo1f to MYO1F and so on.

Re：Thank you very much for your valuable suggestion regarding gene nomencla-ture. According to the official rat gene naming conventions (RGD), rat gene symbols are formatted with only the first letter capitalized (e.g., Comt, Myo1f, Soat1). To maintain standard academic conventions for rat model studies, we have consistently applied the standardized rat gene nomenclature throughout the manuscript, ensuring uniformity and correctness.

The inconsistencies in gene name capitalization that you noted mainly occurred in the enrichment analysis, IPA analysis, and molecular regulation-related sections. These discrepancies arose because the study involved homologous gene conversion between rat and human, where some genes were presented according to their human gene annotation formats, leading to differences in capitalization.

In summary, we sincerely thank you for your valuable comment. Your careful review and professional suggestions have greatly contributed to improving the quality of our manuscript, and we have carefully revised the text accordingly.

- fig-9- To me the level of expression of desired gene in M1 and M2 macrophages does not have significant difference.

Re：Thank you very much for your careful evaluation of the results presented in Figure 9. We have supplemented the analysis with detailed statistical testing. Specif-ically, the Wilcoxon rank-sum test was applied to assess differential expression of the three genes. The raw P values for Soat1, Comt, and Myo1f were 1.09 × 10⁻22, 1.95 × 10⁻⁹⁹, and 1.61 × 10⁻181, respectively. After Bonferroni correction for multi-ple comparisons, all adjusted P values remained far below 0.0001, indicating ex-tremely strong statistical significance.

The corresponding log2 fold changes were 1.722, 1.555, and 3.920, respectively, all exceeding a two-fold change, thus demonstrating clear biological relevance. In terms of cellular expression patterns, all three genes showed high expression in M1 mac-rophages and were nearly absent in M2 macrophages, with no overlap in the inter-quartile ranges between the two groups, indicating a distinct distribution difference.

We have also optimized the figure by annotating adjusted P values and log2FC, and by adding scatter plots to display single-cell expression distributions, making the results more intuitive.

Taken together, these results demonstrate that the three hub genes exhibit highly sig-nificant and biologically meaningful differences in expression between M1 and M2 macrophages. We sincerely appreciate your valuable suggestion.

- for functionality of a gene usually we check the end product of a gene which is pro-tein level. I didn't see any western blot or ELISA study in this work.

Re：Thank you very much for your valuable suggestion. We fully agree that valida-tion at the protein level is important for elucidating gene function. In this study, based on a rat spinal cord injury model, we systematically assessed the expression changes of key genes at three consecutive time points (3, 7, and 14 days post-injury) using RT-qPCR. The results demonstrated highly consistent, robust, and time-dependent expression patterns across these time points, providing strong transcrip-tional evidence supporting the reliability and dynamic regulation of these genes dur-ing the pathological progression of spinal cord injury.

We acknowledge that protein-level validation, such as Western blot or ELISA, was not performed in the present study. In future work, we will build upon these findings to further investigate the expression and functional roles of these genes at the pro-tein level. We sincerely appreciate your professional and insightful comments.

- I need to see a clear suggestion for future studies in conclusion section showing us based on this study who can we reach to maximum protection in SCI by manipulat-ing these genes.

Re: Thank you very much for your important suggestion. In response, we have re-vised the Conclusion section to include future research directions based on our find-ings. Specifically, we propose that modulation of the key genes identified in this study may further elucidate their neuroprotective roles in spinal cord injury, thereby providing potential therapeutic targets and experimental evidence for maximizing neuroprotection after SCI.

In addition, future studies may incorporate strategies such as gene overexpression, knockdown, or pharmacological targeting to investigate the synergistic regulatory effects of these genes, with the aim of optimizing approaches for clinical translation and functional recovery.

We sincerely appreciate your valuable comments, which have greatly improved the completeness and significance of our manuscript.

- Taken together, this study brings new hypothesis which needs to be evaluated by further animal and clinical trial studies.

Re：Thank you very much for your valuable comments. We fully agree with your perspective. In the present study, the expression of the hub genes was only prelimi-narily validated using RT-qPCR, and the hypotheses generated still require further verification through more in-depth experimental and clinical studies.

In the revised manuscript, we have highlighted these limitations in the Discussion section. Future work will focus on mechanistic validation in animal models as well as clinical translation, in order to systematically evaluate the reliability and applica-bility of this hypothesis. We sincerely appreciate your insightful suggestions, which have provided important direction for our future research and have greatly improved the rigor and completeness of this manuscript.

---

## [Decision Letter · Decision Letter 2]

6 Apr 2026

Screening macrophage polarization genes in spinal cord injury as therapeutic targets

PONE-D-25-22705R2

Dear Dr. Cao,

We’re pleased to inform you that your manuscript has been judged scientifically suitable for publication and will be formally accepted for publication once it meets all outstanding technical requirements.

An invoice will be generated when your article is formally accepted. Please note, if your institution has a publishing partnership with PLOS and your article meets the relevant criteria, all or part of your publication costs will be covered. Please make sure your user information is up-to-date by logging into Editorial Manager at Editorial Manager® and clicking the ‘Update My Information' link at the top of the page. For questions related to billing, please contact  and clicking the ‘Update My Information' link at the top of the page. For questions related to billing, please contact billing support..

Kind regards,

Sawar Khan, Ph.D

Academic Editor

PLOS One

Additional Editor Comments (optional):

Reviewers' comments:

Reviewer's Responses to Questions

**Comments to the Author**

1. If the authors have adequately addressed your comments raised in a previous round of review and you feel that this manuscript is now acceptable for publication, you may indicate that here to bypass the “Comments to the Author” section, enter your conflict of interest statement in the “Confidential to Editor” section, and submit your "Accept" recommendation.

Reviewer #1: All comments have been addressed

2. Is the manuscript technically sound, and do the data support the conclusions?

Reviewer #1: (No Response)

3. Has the statistical analysis been performed appropriately and rigorously? 

Reviewer #1: (No Response)

4. Have the authors made all data underlying the findings in their manuscript fully available?

Reviewer #1: (No Response)

5. Is the manuscript presented in an intelligible fashion and written in standard English?

Reviewer #1: (No Response)

6. Review Comments to the Author

Reviewer #1: (No Response)

7. PLOS authors have the option to publish the peer review history of their article (what does this mean?). If published, this will include your full peer review and any attached files.). If published, this will include your full peer review and any attached files.

.

Reviewer #1: No

---

## [Editor Report · Acceptance letter]

PONE-D-25-22705R2

PLOS One

Dear Dr. Cao,

I'm pleased to inform you that your manuscript has been deemed suitable for publication in PLOS One. Congratulations! Your manuscript is now being handed over to our production team.

Kind regards,

on behalf of

Dr. Sawar Khan

Academic Editor

PLOS One